# A Method for Measuring Angular Orientation with Adaptive Compensation of Dynamic Errors

**DOI:** 10.3390/s25164922

**Published:** 2025-08-09

**Authors:** Dimitar Dichev, Iliya Zhelezarov, Borislav Georgiev, Tsanko Karadzhov, Ralitza Dicheva, Hasan Hasanov

**Affiliations:** 1Department of Machine and Precision Engineering, Technical University of Gabrovo, 4 H. Dimitar Str., 5300 Gabrovo, Bulgaria; dichevd@abv.bg (D.D.); izhel@tugab.bg (I.Z.); karadjov_st@abv.bg (T.K.); hasanov77@gmail.com (H.H.); 2Center of Competence “Smart Mechatronic, Eco-and Energy-Saving Systems and Technologies”, Technical University of Gabrovo, 5300 Gabrovo, Bulgaria; 3Department of Precision Engineering and Measurement Instruments, Technical University of Sofia, 8 Kl. Ohridski Blvd, 1000 Sofia, Bulgaria; ralitsa.dichevaa@gmail.com

**Keywords:** angular orientation measurement, MEMS sensors, adaptive Kalman filtration, evaluation of measurement uncertainty in dynamic measurements

## Abstract

This article presents an integrated method for measuring the angular orientation of moving objects, combining a simplified mechanical structure to reduce instrumental errors with a hardware–software platform for adaptive compensation of dynamic errors. Unlike existing approaches, the method avoids inertial element stabilization by using an adaptive Kalman structure for real-time correction. Based on this method, a measuring system for determining roll and pitch has been developed and implemented using a two-channel measurement model with two independent signals and MEMS sensors. The accuracy of the system has been experimentally validated in both static and dynamic modes through a highly accurate reference system with traceability to international standards. A metrologically based methodology for quantitative assessment has also been developed, applying both the theory of error and the theory of uncertainty to provide an objective, reproducible, and traceable evaluation under real-world conditions.

## 1. Introduction

Measurements performed under dynamic conditions present significantly greater challenges compared to static measurements. The results of such measurements are time-dependent, and although time itself can be measured with reasonable accuracy, various factors—such as inertial forces and moments, nonlinear effects arising from motion dynamics, systematic sensor errors, environmental disturbances, and limitations of mathematical models—introduce considerable obstacles to achieving reliable real-time results [1,2,3,4,5]. This necessitates the adoption of a systems-level approach that integrates modeling with adaptive uncertainty management [6,7].

In this context, methods and systems for estimating the angular orientation of moving objects merit close examination. Current theoretical and practical frameworks classify orientation estimation technologies into three major directions, reflecting the leading trends in their evolution. The first direction pertains to advancements in sensor technologies, which form the foundational components of contemporary measurement systems. This includes both enhancements to existing sensors and the development of new generations of sensing elements with improved accuracy, robustness, and resistance to external influences.

In recent years, MEMS sensors have emerged as key technologies for inertial measurements due to their compact form factor and low cost. Modern developments include frequency-modulated gyroscopes, resonant and optical accelerometers, and thermally stabilized structures [8,9,10,11,12,13,14]. These solutions exhibit enhanced thermal stability, broader frequency bandwidth, and improved dynamic accuracy—critical properties for precise measurements in dynamic scenarios. MEMS sensors can be categorized based on the measured variable, underlying physical principle (capacitive, piezoresistive, thermal, resonant, and optical), and behavior under inertial loads. Among them, capacitive and resonant sensors offer an optimal balance between sensitivity, stability, and dynamic accuracy, making them preferable for real-time applications [15,16]. In contrast, piezoresistive and thermal sensors demonstrate limited stability under dynamic conditions [17,18,19], while optical sensors demand more complex integration schemes [20,21]. Beyond MEMS technologies, contemporary systems also utilize electrical, optical, laser-based, and quantum sensors, expanding capabilities for accurate measurements in diverse and extreme environments [22,23,24,25,26,27].

The second major direction involves the design of integrated measurement platforms that combine data from multiple sensor types—MEMS accelerometers, gyroscopes, optical systems, and GPS receivers. Such sensor fusion improves measurement reliability and accuracy by compensating for the individual limitations of each technology [28,29,30]. Common examples include inertial measurement units (IMUs) and attitude and heading reference systems (AHRSs), which synthesize information from MEMS sensors to estimate roll, pitch, and yaw [31,32,33,34,35,36]. One illustrative solution is the LARA system [37], which integrates accelerometers and gyroscopes to improve accuracy; however, its performance under dynamic loads remains uncharacterized. These systems serve as the foundation for multi-sensor architectures featuring enhanced resilience and precision.

The third trend focuses on the development of adaptive data processing algorithms that address the compensation of dynamic errors—a primary source of cumulative inaccuracies in inertial measurement systems. Advanced implementations of Kalman filtering, including Unscented Kalman Filter (UKF), Cubature Kalman Filter (CKF), and Particle Filter (PF), enable adaptation to nonlinear dynamics and noisy environments [38,39,40,41].

Despite the advancements in measurement systems discussed above, numerous challenges remain. These include cumulative error accumulation over time, diminished performance under adverse external conditions, complexity in multi-technology integration, and stringent calibration requirements. Certain systems are sensitive to environmental disturbances such as vibrations, mechanical shocks, or magnetic interference, reducing their reliability in dynamic settings. Moreover, many applications require accurate orientation measurements under both dynamic and static conditions—for instance, when specifying roll and pitch parameters during cargo loading on marine vessels. To achieve high dynamic accuracy, complex measurement systems are often deployed, which introduce significant instrumental errors. Such errors may lead to substantial deviations in the determination of critical parameters, directly impacting operational safety and efficiency. Additionally, most existing systems lack a well-defined methodology for dynamic calibration, complicating traceability and real-time assessment of measurement uncertainty. This aspect is particularly crucial in applications where system reliability depends on accurately assessed dynamic characteristics.

These limitations hinder the realization of precise and dependable measurements in both dynamic and static operational modes—an essential requirement for orientation control of mobile objects. Therefore, this work proposes a measurement method that simultaneously offers high accuracy in dynamic conditions and minimal instrumental error under static conditions. Based on this method, a specific measurement system has been designed, incorporating three adaptive mechanisms: estimation of measurement errors, compensation of model imperfections, and correction of systematic sensor deviations. These mechanisms are implemented within a Kalman filter structure using time-varying parameters, estimated adaptively at each algorithmic step based solely on current measurements—eliminating the need for external computational modules or supplementary channels. The proposed method provides a framework for developing other system-level architectures tailored to specific application requirements.

To facilitate the deployment of the method in real-world scenarios, an accuracy assessment methodology for dynamic operation has also been developed. This approach incorporates principles from error theory and uncertainty analysis, accounting for the influence of sensors, mechanical architecture, processing algorithms, and external factors. It ensures traceability through calibration against international standards.

## 2. Description of the Measurement Method and System for Measuring Roll and Pitch

The proposed method for measuring the parameters of moving objects combines, on one hand, a simplified metrological structure of the mechanical modules of the measurement systems to reduce instrumental errors and improve reliability, and on the other hand, it ensures the necessary dynamic accuracy by correcting dynamic errors in real-time. Unlike most existing measuring instruments in this field, which rely on the stabilization of inertial elements (e.g., through the gyroscopic effect), the proposed method employs a simplified mechanical structure with a physical pendulum. This structure provides high reliability and low instrumental error under static conditions, such as during initial orientation adjustments, for example, when loading vessels. Dynamic accuracy is achieved through algorithmic data processing that compensates for errors caused by pendulum deflection under inertial influences. In this way, the measuring system operates through an integrated method that combines a mechanical sensing element, two independent measuring channels with MEMS sensors, and an advanced Kalman filter for real-time dynamic error correction. This concept allows for the adaptation of measurement systems to complex dynamic conditions without the need for expensive stabilization mechanisms. The method is illustrated by a system for measuring the roll and pitch of a moving object.

The mechanical module of the measurement system is presented in Figure 1 and Figure 2. It consists of a device body 1 (Figure 1), which is securely attached to the moving object. The physical pendulum 2 (Figure 1), which has two degrees of freedom, is mounted to the housing via a cardan joint 3 (Figure 2). The pendulum serves as a reference for the vertical position and provides a mechanical realization of verticality, with its two degrees of freedom enabling the measurement of roll and pitch angles.

Along the axes of each degree of freedom of the pendulum, photoelectric Absolute Rotary Encoders 4 (Figure 2) are positioned to register the angles between the pendulum and the device body for each degree of freedom. The encoders used are model FKP-13A-18, manufactured by ZGPU GRUP (ZGPU GRUP, Gabrovo, Bulgaria). The two photoelectric Absolute Encoders (AEs) are of the same type and operate by differentially parallel scanning each digit of the rotating scale, encoded in Gray code, which eliminates interference errors and provides a wide operating temperature range. The encoders have a base resolution of 2^13^ bits, which is increased to 2^18^ bits using an additional digit, analogous to the least significant bit but out of phase by 90 electrical degrees, and combining the application of analog and digital interpolation using advanced signal processing algorithms. They are characterized by high accuracy, high noise immunity, fast response, a wide supply voltage range, and compact dimensions, making them suitable for application in the current system. This allows for the accurate determination of the angles of rotation of the moving object (roll and pitch) relative to the vertical, thus enabling a simple and reliable measurement method. It is important to note that this approach is effective only when the pendulum remains pointed vertically downward. In the event of dynamic impacts on the object, the pendulum may temporarily deviate from the vertical, resulting in dynamic errors. Therefore, the proposed method is structured so that, through a hardware-algorithmic platform, it corrects the results in real-time by removing these errors. Unlike most methods that stabilize measurement systems to eliminate the influence of inertial impacts, this method does not rely on complex and expensive stabilization systems but instead provides correction of errors caused by these impacts.

The structural diagram of the measurement system is shown in Figure 3. It consists of two parallel circuits, a measuring circuit and a correction circuit, which operate in tandem to ensure accuracy under dynamic conditions.

The measuring circuit is the main functional component of the system, providing initial data collection on the angles of roll and pitch. It uses a photoelectric AE connected to the pendulum to provide information about the relative angles between the pendulum and the instrument body. These signals reflect the dynamics of the object’s motion and form the basis for the measurement results. However, the measurements can be affected by external inertial forces and moments that cause the pendulum to deviate from the vertical. Therefore, additional correction is necessary through the correction circuit.

The correction circuit consists of two parallel correction subsystems that measure the pendulum’s deviation from the vertical and provide corrections for dynamic errors. The first correction subsystem employs a scheme in which AHRS 1 (item 1 in Figure 2) is mounted directly on the pendulum. The AHRS measures the angles of deviation of the pendulum from the vertical (angles α and β). These signals are used to determine the pendulum deflection, which is then factored into correcting the measurements from the photoelectric AE. The second correction subsystem includes another AHRS 2 of the same type (position 2 in Figure 2), mounted on the device’s body and fixed relative to the moving object. This AHRS measures the dynamic movements of the object itself. By subtracting the signal from the second AHRS from the signal from the photoelectric AE, the angle of the pendulum’s angle of deviation from the vertical is obtained. This allows for additional confirmation and the development of more precise measurement correction algorithms, with the two correction subsystems providing accurate measurements in the presence of external forces and moments that may affect the sensors. The signals received from the correction subsystems provide independent measurements of the pendulum’s deviation from the vertical. Since both AHRS sensors are subjected to similar external forces and moments, the system can compare their readings and eliminate common errors. An AHRS sensor, model WT901SDCL-BT50, is used in both subsystems. It features a three-dimensional gyroscope, accelerometer, and magnetometer, along with a built-in Kalman filter and wireless communication capability. The model was selected for its compact design, high level of integration, and reliable performance under dynamic conditions.

In addition, the system uses an advanced Kalman filter, based on the dynamic model of the pendulum, allowing for real-time processing of the measured signals. The filter optimizes data from MEMS sensors and corrects errors related to inertial influences and the dynamics of object movement. This enhances measurement precision and high reliability, even under complex dynamic conditions. To further improve the reliability and accuracy of the system, the pendulum readings are used to correct the data received from the MEMS sensors in the AHRS. When the pendulum is positioned vertically, the system checks whether the angles read by the AHRS match those from the photoelectric AE. If there is a discrepancy between the values, a correction is made. Since the pendulum offers greater reliability during extended periods, such as on ships on long voyages at sea, this methodology significantly enhances the overall reliability and accuracy of the measurement system.

### Derivation of the Dynamic Model Determining the Angular Displacement of the Pendulum from the Vertical

The derivation of the dynamic model describing the movement of the pendulum relative to the vertical axis is based on an analysis of the coordinate systems and the rotational movements of the moving object presented in Figure 4. For this purpose, the terrestrial coordinate system OXYZ is used, relative to which the angular and translational movements of the object are determined. The coordinate system $OX_{2}Y_{2}Z_{2}$, is related to the moving object, with the *OZ*_2_ axis pointing vertically downwards when the roll and pitch of the object are equal to zero. The *OY*_2_ axis is oriented along the longitudinal axis of the moving object, for example, towards the bow, while the *OX*_2_ axis is directed transversely.

During movement, the axes of the system $OX_{2}Y_{2}Z_{2}$ rotate relative to the Earth’s coordinate system, and these rotations are described by the angles of roll *θ* and pitch *ψ*. To determine the orientation of the moving object relative to the vertical, it is necessary to measure the current values of the angles *θ* and *ψ*.

The center of suspension of the pendulum $O_{1}$ is located at a distance defined by the coordinates *d_x_*, *d_y_*, *d_z_* from the center of gravity of the moving object. For a more accurate and consistent formulation of the model shown in Figure 4, both the terrestrial coordinate system *OXYZ* and the coordinate system associated with the moving object $OX_{2}Y_{2}Z_{2}$, are translationally shifted to point $O_{1}$. The axes of the new coordinate systems $O_{1}X^{\prime }Y^{\prime }Z^{\prime }$ and $O_{1}X_{2}^{\prime }Y_{2}^{\prime }Z_{2}^{\prime }$ remain parallel to those of the original systems *OXYZ* and $OX_{2}Y_{2}Z_{2}$. Figure 4 also shows an intermediate coordinate system $O_{1}X_{1}^{\prime }Y_{1}^{\prime }Z_{1}^{\prime }$, which determines the angular displacement of the system associated with the object $O_{1}X_{2}^{\prime }Y_{2}^{\prime }Z_{2}^{\prime }$ about the axis $O_{1}X^{\prime }$ at an angle *θ*, which describes the roll of the object. The angular displacement about the $O_{1}Y_{1}^{\prime }$ axis describes the object’s pitch by the angle *ψ*. This rotation, together with the displacement along the roll *θ*, determines the final angular orientation of the $X_{2}Y_{2}Z_{2}$ system relative to the Earth coordinate system.

The pendulum is connected to the coordinate system $O_{1}xyz$, whose center coincides with the suspension point $O_{1}$. The axis $O_{1}z$ is oriented vertically, aligned with the direction of the pendulum, while the axes $O_{1}x$ and $O_{1}y$ are perpendicular to it, corresponding to the transverse and longitudinal directions, respectively. The inertial forces and moments generated by the motion of the object induce corresponding linear and angular accelerations, which lead to angular deviations of the pendulum from the vertical. The linear accelerations at the suspension point include the transverse-horizontal acceleration ${\ddot{\eta }}_{o}$ and the longitudinal-horizontal acceleration ${\ddot{\xi }}_{o}$. Together with them, the angular accelerations along the roll $\ddot{\theta }$ and pitch $\ddot{\psi }$ also have an influence on the dynamics of the system.

Although the moving object has six degrees of freedom, the inertial forces and moments acting on the pendulum are primarily generated by the linear and angular accelerations described above. This is because the pendulum itself has only two degrees of freedom, and its axes are oriented in such a way that these specific accelerations directly influence the dynamic behavior of the system.

The displacements of the pendulum from the vertical position are described by the angles α and β, which represent errors caused by the pendulum’s deviation along the roll and pitch axes, respectively. These angles define the dynamic deviation of the pendulum and result in a discrepancy between the actual roll and pitch *θ* and *ψ*, and the values registered by the photoelectric AE, which will register the total values of the angles, i.e., θ+α and ψ+β. Thus, the errors in the measurement of the roll and pitch are determined by the angles *α* and *β*, which are in practice functions of time $\alpha \left(t\right)$ and $\beta \left(t\right)$, because of which they take on the characteristics of dynamic errors.

The differential equations defining the time variation in the dynamic errors $\alpha \left(t\right)$ and $\beta \left(t\right)$ are derived from the geometric, kinematic, and dynamic relations established by analysis of Figure 4. To accurately determine the characteristics of the dynamic error, the equations are compiled based on geometric relations describing the deviations of the pendulum from the vertical. The Lagrange principle is applied, considering the system’s kinetic and potential energy. The equations include the main factors influencing the dynamics of the system: angular and linear accelerations, inertial forces, gravity, and dynamic impacts generated by the movement of the object, and they have the following form:(1)$$\ddot{\alpha }+\frac{b_{1}}{I_{y}}\cdot \dot{\alpha }+\frac{\left(m\cdot g\cdot l+m\cdot {\ddot{\zeta }}_{o}\cdot l\right)}{I_{y}}\cdot \alpha =\frac{m\cdot l}{I_{y}}\cdot {\ddot{\eta }}_{o}-\frac{I_{y}+m\cdot l\cdot d_{z}}{I_{y}}\cdot \ddot{\theta };\;\ddot{\beta }+\frac{b_{2}}{I_{x}}\cdot \dot{\beta }+\frac{\left(m\cdot g\cdot l+m\cdot {\ddot{\zeta }}_{o}\cdot l\right)}{I_{x}}\cdot \beta =\frac{m\cdot l}{I_{x}}\cdot {\ddot{\xi }}_{o}-\frac{I_{x}+m\cdot l\cdot d_{z}}{I_{x}}\cdot \ddot{\psi }$$
where m—the mass of the pendulum; $I_{x}$, $I_{y}$—moments of inertia of the pendulum with respect to the axes $O_{1}y$ and $O_{1}x$, respectively; g—the gravitational acceleration; *l*—the length of the pendulum; and ${\ddot{\eta }}_{o}$, ${\ddot{\xi }}_{o}$, ${\ddot{\zeta }}_{o}$—the longitudinal-horizontal linear acceleration, the transverse-horizontal linear acceleration, and the vertical linear acceleration of the object, respectively.

## 3. Modeling and Tuning of the Elements of the Kalman Filter

### 3.1. Configuring the Basic Elements of the Kalman Filter

The advanced Kalman filter developed for measuring the angular orientation of a moving object relies on two main components: the dynamic model of the pendulum and the measurement model. The dynamic model describes the behavior of the sensing element under the influence of inertial forces and gravity, while the measurement model combines two independent signals to provide an optimal estimate of the pendulum’s deviation from the vertical. The Kalman filter integrates these signals to minimize noise and external disturbances. The structure of the Kalman filter algorithm is presented in Figure 5.

The state of the system is described by the angles of deviation of the pendulum from the vertical, α(t) and β(t), and their corresponding angular velocities and accelerations $\dot{\alpha }\left(t\right)$, $\ddot{\alpha }\left(t\right)$, and $\dot{\beta }(t)$, $\ddot{\beta }(t)$. As noted, these parameters determine the characteristics of the dynamic error in measuring the orientation of the moving object.

The identification and correction of these errors in the data obtained from the photoelectric encoders form an essential part of the proposed method for the actual orientation parameters θ(t) and ψ(t). By using the differential Equation (1), which represents the deviations of the pendulum from the vertical as a function of time, the dynamic model of the system is integrated in the following well-established form, expressed in vector-matrix form [42]:(2)$$\dot{\chi }\left(t\right)=\Phi \cdot \chi \left(t\right)+\Omega \cdot \upsilon \left(t\right)+w\left(t\right)$$
where$\chi \left(t\right)={\left[\begin{array}{cccc} \alpha \left(t\right) & \dot{\alpha }\left(t\right) & \beta \left(t\right) & \dot{\beta }\left(t\right) \end{array}\right]}^{T}$ is the state vector;Φ is the state transition matrix defining the internal dynamics:



(3)
$$\Phi =\left[\begin{array}{cccc} 0 & 1 & 0 & 0\\ -\frac{mgl}{I_{y}} & -\frac{b_{1}}{I_{y}} & 0 & 0\\ 0 & 0 & 0 & 1\\ 0 & 0 & -\frac{mgl}{I_{x}} & -\frac{b_{2}}{I_{x}} \end{array}\right]$$




$\upsilon \left(t\right)={\left[\begin{array}{ccccccc} \dot{\theta }\left(t\right) & \ddot{\theta }\left(t\right) & \dot{\psi }\left(t\right) & \ddot{\psi }\left(t\right) & {\ddot{\eta }}_{o}\left(t\right) & {\ddot{\xi }}_{o}\left(t\right) & {\ddot{\zeta }}_{o}\left(t\right) \end{array}\right]}^{T}$ is the vector of control impacts, the current values of which are measured by the AHRS mounted on the system body;Ω is the matrix that connects the controlling influences on the state:




(4)
$$\Omega =\left[\begin{array}{ccccccc} 0 & 0 & 0 & 0 & 0 & 0 & 0\\ 0 & -\frac{I_{y}+mld_{z}}{I_{y}} & 0 & 0 & \frac{ml}{I_{y}} & 0 & -\frac{ml}{I_{y}}\\ 0 & 0 & 0 & 0 & 0 & 0 & 0\\ 0 & 0 & 0 & -\frac{I_{x}+mld_{z}}{I_{x}} & 0 & -\frac{ml}{I_{x}} & -\frac{ml}{I_{x}} \end{array}\right]$$




$w\left(t\right)$ represents stochastic noise in the process, which reflects dynamic indeterminacies and unpredictable influences on the system, and is assumed as a random magnitude with zero meaning and a covariance matrix Q, which defines the dispersion of this noise.


Since the filter operates in an iteration structure, it is necessary to transform the predictive model from continuous time *t* to discrete steps *k*, which reflect the successive iterations of the algorithm. The predictive equations for each step k are represented by the following well-known formulation [42]:(5)$$\chi_{k|k-1}=\Phi \cdot \chi_{k-1}+\Omega \cdot \upsilon_{k-1}$$
where $\chi_{k|k-1}$ is the prediction for the state of the system at step k, based on the data from the previous step *k* − 1.

The covariance matrix of the prediction error is updated as follows:(6)$$P_{k|k-1}=\Phi \cdot P_{k-1}\cdot \Phi^{T}+Q_{k-1}$$
where the matrix $P_{k|k-1}$ reflects the indeterminacy in the state prediction and includes the influence of the noise process covariance described by the matrix $Q_{k-1}$.

It is considered in (6) that, according to the adaptive algorithm adopted in this work for the matrix Q, it changes at each step of the iteration procedure, which is why it was written in the form $Q_{k-1}$.

Measurements in dynamic mode are significantly more complex than those in static mode. Under dynamic conditions, time (t) and its derivatives—velocities and accelerations—play a crucial role, leading to the emergence of inertial forces and moments. These dynamic factors can cause deviations and impact the accuracy of the measuring instruments. This is why the measurement model in the system is designed as a two-channel system, utilizing two independent signals to achieve maximum accuracy in assessing the dynamic deviation of the pendulum from the vertical. From a metrological perspective, different measurement channels and a greater number of independent measurements enhance statistical reliability, resulting in a more precise final assessment. The first signal is received from the AHRS mounted on the pendulum, which provides direct information about the angles of deviation α and β. This signal is essential for the real-time assessment of the pendulum’s deviation from the vertical. However, because of its position on the pendulum, the signal is susceptible to inertial noise and external disturbances caused by additional movements, which also generate inertial forces and moments.

To reduce the influence of these impacts, the measurement model includes a second measurement circuit, the signal of which is formed as the difference between the readings of the photoelectric converter and the AHRS mounted on the hull. This circuit includes a second AHRS, which is not affected by the oscillations of the pendulum and ensures the independence of the measurement from the additional inertial forces and moments acting on the pendulum. Thus, the common metrological circuit, including both AHRS and the photoelectric encoder, creates conditions for the statistical correlation of the data from the two measurement channels, which allows for improved measurement accuracy in the dynamic mode of operation.

Along with this, this measurement structure aims to include all measurement tools in the system, enabling the construction of a more accurate reference signal in real time. This signal serves as the basis for modeling and calculating the covariance matrix of the dispersion of measurement errors, which encompasses the accuracy of each component and forms the total accuracy of the system. By integrating the data and metrological characteristics of the individual measurement channels, improved system accuracy is achieved in the dynamic mode of operation.

The larger number of independent measurements increases the accuracy of the estimation and the robustness of the measurement process. The use of two independent measurement channels for each of the quantities α and β not only increases the reliability of the estimation but also provides better handling of random errors through Kalman estimation based on the covariance matrices. This two-channel measurement structure provides increased sustainability of the estimations and reduces the total metrological uncertainty, thus improving accuracy and contributing to the stability of the system.

In this regard, the measurement model is expressed as(7)$$z\left(t\right)=\left[\begin{array}{c} z_{1\;}\left(t\right)\\ z_{2}\left(t\right) \end{array}\right]=\left[\begin{array}{c} H_{1}\\ H_{2} \end{array}\right]\cdot \chi \left(t\right)+\left[\begin{array}{c} \vartheta_{1}\left(t\right)\\ \vartheta_{2}\left(t\right) \end{array}\right]$$
where $z_{1}\left(t\right)$ and $z_{2}\left(t\right)$ are the measurements from the two independent channels in the system and contain information about the deviation angles *α* and *β* at the current time *t*; $H_{1}$ and $H_{2}$ are the measurement matrices corresponding to the two channels, structured to extract the values of the corresponding variables *α* and *β* from the state vector $\chi \left(t\right)$; and $\vartheta_{1}\left(t\right)$ and $\vartheta_{2}\left(t\right)$ are the noise and error vectors in the measurements for each of the two channels, reflecting random errors and disturbances that accompany the measurement process.

$H_{1}$ and $H_{2}$ are matrices of the following type:(8)$$H_{1,2}=\left[\begin{array}{cccc} 1 & 0 & 0 & 0\\ 0 & 0 & 1 & 0 \end{array}\right]$$

Accordingly, Equation (7), reducing to its discrete form for the current iteration step k, takes the following form:(9)$$z_{k}=\left[\begin{array}{c} z_{1_{k}}\\ z_{2_{k}} \end{array}\right]=\left[\begin{array}{c} \alpha_{k}^{z_{1}}\\ \beta_{k}^{z_{1}}\\ \alpha_{k}^{z_{2}}\\ \beta_{k}^{z_{2}} \end{array}\right]=\left[\begin{array}{cccc} 1 & 0 & 0 & 0\\ 0 & 0 & 1 & 0\\ 1 & 0 & 0 & 0\\ 0 & 0 & 1 & 0 \end{array}\right]\cdot \left[\begin{array}{c} \alpha_{k}\\ {\dot{\alpha }}_{k}\\ \beta_{k}\\ {\dot{\beta }}_{k} \end{array}\right]+\left[\begin{array}{c} \vartheta_{1_{k}}^{\alpha }\\ \vartheta_{1_{k}}^{\beta }\\ \vartheta_{2_{k}}^{\alpha }\\ \vartheta_{2_{k}}^{\beta } \end{array}\right]$$
where $\alpha_{k}^{z_{1}}$ and $\beta_{k}^{z_{1}}$ are the measured values of the pendulum deviation angles from the vertical from the first measurement channel in the current step *k*; $\alpha_{k}^{z_{2}}$ and $\beta_{k}^{z_{2}}$ are the measured values of the pendulum deviation angles from the vertical from the second measurement channel in the current step *k*; $\vartheta_{1_{k}}^{\alpha }$ and $\vartheta_{1_{k}}^{\beta }$ are the noises associated with the measurement *α* and *β* in the first measurement channel for the current step k; and $\vartheta_{2_{k}}^{\alpha }$ and $\vartheta_{2_{k}}^{\beta }$ are the noises associated with the measurement *α* and *β* in the second measurement channel for the current step *k*.

Based on the measurement model defined above, the prediction deviation $y_{k}$ is defined as the difference between the values of the vector of actual measurement $z_{k}$ and the predicted measurement $H\cdot \chi_{k|k-1}$, based on the state prediction in step *k*:(10)$$y_{k}=z_{k}-H\cdot \chi_{k|k-1}$$

The deployed matrix form of Equation (10) will have the following type:(11)$$y_{k}=\left[\begin{array}{c} z_{1_{k}}\\ z_{2_{k}} \end{array}\right]-\left[\begin{array}{cccc} 1 & 0 & 0 & 0\\ 0 & 0 & 1 & 0\\ 1 & 0 & 0 & 0\\ 0 & 0 & 1 & 0 \end{array}\right]\cdot \left[\begin{array}{c} \alpha_{k|k-1}\\ {\dot{\alpha }}_{k|k-1}\\ \beta_{k|k-1}\\ {\dot{\beta }}_{k|k-1} \end{array}\right]$$
where $y_{k}$ is the following four-component vector:(12)$$y_{k}=\left[\begin{array}{c} y_{1_{k}}^{\alpha }\\ y_{1_{k}}^{\beta }\\ y_{2_{k}}^{\alpha }\\ y_{2_{k}}^{\beta } \end{array}\right]=\left[\begin{array}{c} \alpha_{k}^{z_{1}}-\alpha_{k|k-1}\\ \beta_{k}^{z_{1}}-\beta_{k|k-1}\\ \alpha_{k}^{z_{2}}-\alpha_{k|k-1}\\ \beta_{k}^{z_{2}}-\beta_{k|k-1} \end{array}\right]$$

The state estimate, which provides the final result of the Kalman filter output for the angular deviations and their derivatives, is obtained by correcting the prediction with the deviation $y_{k}$ according to the following equation:(13)$${\hat{\chi }}_{k}=\left[\begin{array}{c} {\hat{\alpha }}_{k}\\ {\hat{\dot{\alpha }}}_{k}\\ {\hat{\beta }}_{k}\\ {\hat{\dot{\beta }}}_{k} \end{array}\right]=\chi_{k|k-1}+K_{k}\cdot y_{k}$$
where $K_{k}$ is the Kalman coefficient.

### 3.2. Configuring the Main Elements of the Correction Matrix

To enhance the reliability and accuracy of the measurement system, particularly for applications involving the continuous movement of vehicles, such as ships, a method for correcting systematic errors has been developed based on an additional algorithm. This algorithm uses the positional properties of the physical pendulum as a reference element to determine corrections to the AHRS readings, which are based on data from inertial sensors. The ability of the pendulum to assume a vertical position under the influence of gravity provides a reliable means for determining systematic errors in the AHRS, arising from drifts, temperature influences, and external factors.

The concept focuses on identifying systematic errors at the moment when the pendulum settles into the vertical position. This position is defined by processing data from photoelectric AE, ensuring it remains constant within a specific time interval. The systematic errors are calculated as the difference between the current readings of the two AHRSs and the reference zero deviation, with the pendulum serving as a reference element in the Kalman filter algorithm. These errors are integrated into the algorithm for subsequent correction of the measurements, resulting in a more accurate assessment of the system’s state.

The most effective approach for adaptive integration in the Kalman filter structure is the correction of systematic errors due to the change in the zero value of inertial measurement systems, to be performed by modifying Equation (10), defining the prediction deviation $y_{k}$, that is(14)$$y_{k}=C_{k}\cdot z_{k}-H\cdot \chi_{k|k-1}$$
where $C_{k}$ is the correction matrix.

The matrix $C_{k}$ has the following structure:(15)$$C_{k}=\left[\begin{array}{cccc} 1-\delta \alpha_{{sys}_{k}}^{{AHRS}_{1}} & 0 & 0 & 0\\ 0 & 1-\delta \beta_{{sys}_{k}}^{{AHRS}_{1}} & 0 & 0\\ 0 & 0 & 1-\delta \alpha_{{sys}_{k}}^{{AHRS}_{2}} & 0\\ 0 & 0 & 0 & 1-\delta \beta_{{sys}_{k}}^{{AHRS}_{2}} \end{array}\right]$$
where $1-\delta \alpha_{{sys}_{k}}^{{AHRS}_{1}},$ $1-\delta \beta_{{sys}_{k}}^{{AHRS}_{1}}$, $1-\delta \alpha_{{sys}_{k}}^{{AHRS}_{2}}$, $1-\delta \beta_{{sys}_{k}}^{{AHRS}_{2}}$ are the values of systematic errors determined at a time point actual to the step k.

The determination of the time interval during which it can be guaranteed that the pendulum is in a vertical position is based on a combination of the dynamic characteristics of the physical pendulum and an analysis of noise in the measurement channels. A key consideration is the natural oscillation frequency of the pendulum, which defines the time scale of its inherent motion.

Based on the simulation results, it was found that the time interval should cover at least three full periods of oscillation to eliminate the influence of transient effects and short-term disturbances. Additionally, this interval must account for the noise levels in the photoelectric AE and be selected such that variations in their readings do not exceed the resolution (discreteness) of the measuring devices within the given period. Experimental studies and simulations conducted showed that the time interval for reliable identification of the vertical position in the specific prototype is 3 s.

### 3.3. Setting Up the Matrix Defining the Variances of Errors in the Measurements R

The matrix **R** defines the variances of errors in the measurements, determining the influence of random noise and other uncertainties in the measurement channels on the final state estimate. This matrix is directly related to the measurement noise components $\vartheta_{1}\left(t\right)$ and $\vartheta_{2}\left(t\right)$, which are assumed to be independent random magnitudes. These noises represent part of the total uncertainty in the measurements of the two channels, associated with the random errors that affect the final estimate. The inclusion of these values in the model is necessary to account for the real influence of random errors and to reduce the effects of the uncertainty inherent in the measuring means.

The structure and values of the matrix **R** are determined through an analysis of the noise characteristics in the measurement channels. This ensures optimal weighting of the measurement data and minimizes the total error in the state assessment. The general form of the matrix **R** is(16)$$R=\left[\begin{array}{cccc} \sigma_{\alpha^{z_{1}}}^{2} & 0 & 0 & 0\\ 0 & \sigma_{\beta^{z_{1}}}^{2} & 0 & 0\\ 0 & 0 & \sigma_{\alpha^{z_{2}}}^{2} & 0\\ 0 & 0 & 0 & \sigma_{\beta^{z_{2}}}^{2} \end{array}\right]$$
where the elements $\sigma_{\alpha^{z_{1}}}^{2},\sigma_{\beta^{z_{1}}}^{2},\sigma_{\alpha^{z_{2}}}^{2},\sigma_{\beta^{z_{2}}}^{2}$ represent the dispersions of the measurement errors for the corresponding variables in the channels $z_{1}$ and $z_{2}$.

The diagonal structure of Matrix (16) is justified by the assumption that the noise in the measurement channels is uncorrelated, meaning that the correlation between the errors in the different channels can be neglected. This configuration of R is based on the characteristics of the measurement signals in z(t) and is consistent with the nature of the independent measurements for the angles α and β in each channel.

The determination of the dispersions in the **R** matrix is based on analysis of the random errors occurring during measurements in each channel. From a metrological point of view, these errors are defined as differences between the measured value and the corresponding reference value, which serves as an absolute basis for assessing the accuracy of measurements. In cases where there is no direct reference quantity, it is necessary to establish a reference quantity that provides the closest approximation to the ideal reference. This reference quantity is determined by integrating data from all measuring devices in the system, taking into account the dynamic operating conditions. This approach ensures maximum proximity to the ideal reference value. The reference quantity serves as a basis for assessing random errors in measurements and aligns with the characteristics of the system’s dynamic operating mode.

When constructing the reference quantity for estimating random errors in measurements, the least squares method was adopted due to its ability to combine data from all measuring devices in the system and provide the most accurate estimate of the actual value at the current moment. This approach is particularly well-suited for systems with periodic or smoothly varying movements—such as those seen in a physical pendulum—since it offers rapid responsiveness to changes in the measured values while minimizing the influence of random noise.

To determine the values of the dispersions $\sigma_{\alpha^{z_{1}}}^{2},\sigma_{\beta^{z_{1}}}^{2},\sigma_{\alpha^{z_{2}}}^{2},\sigma_{\beta^{z_{2}}}^{2}$, the measurement data are recorded for each step *i* from k−M+1 to k in two matrices, respectively, $\rho_{\alpha }$ for $\alpha_{i}^{z_{1}}$ and $\alpha_{i}^{z_{2}}$, and $\rho_{\beta }$ for $\beta_{i}^{z_{1}}$ and $\beta_{i}^{z_{2}}$, each of which is combined into a single vector, i.e.,(17)$$\rho_{\alpha }=\left[\begin{array}{c} \alpha_{k-M+1}^{z_{1}}\\ \alpha_{k-M+1}^{z_{2}}\\ \alpha_{k-M+2}^{z_{1}}\\ \alpha_{k-M+2}^{z_{2}}\\ \vdots \\ \alpha_{k}^{z_{1}}\\ \alpha_{k}^{z_{2}} \end{array}\right];\;\rho_{\beta }=\left[\begin{array}{c} \beta_{k-M+1}^{z_{1}}\\ \beta_{k-M+1}^{z_{2}}\\ \beta_{k-M+2}^{z_{1}}\\ \beta_{k-M+2}^{z_{2}}\\ \vdots \\ \beta_{k}^{z_{1}}\\ \beta_{k}^{z_{2}} \end{array}\right]\;$$
where $\rho_{\alpha }$ and $\rho_{\beta }$ are vectors of dimension 2M×1.

The matrix equation used to compute the coefficients of the least squares curve at step *k* is [43,44]:(18)$$a_{k}={\left(X_{k}^{T}\cdot X_{k}\right)}^{-1}\cdot X_{k}^{T}\cdot \rho$$
where the matrix of regressors $X_{k}$ in step *k* will be of dimension 2·M×3, i.e.,(19)$$X_{k}=\left[\begin{array}{ccc} {\left(k-M+1\right)}^{2} & \left(k-M+1\right) & 1\\ {\left(k-M+1\right)}^{2} & \left(k-M+1\right) & 1\\ {\left(k-M+2\right)}^{2} & \left(k-M+2\right) & 1\\ {\left(k-M+2\right)}^{2} & \left(k-M+2\right) & 1\\ \vdots & \vdots & \vdots \\ k^{2} & k & 1\\ k^{2} & k & 1 \end{array}\right]$$

The dispersions $\sigma_{\alpha^{z_{1}}}^{2},\sigma_{\beta^{z_{1}}}^{2},\sigma_{\alpha^{z_{2}}}^{2},\sigma_{\beta^{z_{2}}}^{2}$ in step *k* are then calculated using the following formulas:(20)$$\sigma_{\alpha^{z_{1}}}^{2}=\frac{1}{M-1}\cdot \sum_{i=k-M+1}^{k}{\left(\alpha_{i}^{z_{1}}-\alpha_{i}^{ref}\right)}^{2};\;\sigma_{\alpha^{z_{2}}}^{2}=\frac{1}{M-1}\cdot \sum_{i=k-M+1}^{k}{\left(\alpha_{i}^{z_{2}}-\alpha_{i}^{ref}\right)}^{2};\;\sigma_{\beta^{z_{1}}}^{2}=\frac{1}{M-1}\cdot \sum_{i=k-M+1}^{k}{\left(\beta_{i}^{z_{1}}-\beta_{i}^{ref}\right)}^{2};\;\sigma_{\beta^{z_{2}}}^{2}=\frac{1}{M-1}\cdot \sum_{i=k-M+1}^{k}{\left(\beta_{i}^{z_{2}}-\beta_{i}^{ref}\right)}^{2}$$
where $\alpha_{i}^{ref}$ and $\beta_{i}^{ref}$ are the current values of *α* and *β* at step *i*, determined by the least squares method.

An approach based on the adaptation of the derivative (the rate of change) of the relevant measurement signal was used to determine the value of *M*. If the signal changes rapidly, this may be an indication of significant dynamics requiring less *M*, but if the signal changes slowly, *M* may increase, i.e.,(21)$$M_{k+1}=\left\{\begin{array}{c} M_{k}-1,if\;\left|\frac{d\alpha_{k}}{dt}\right|>\epsilon_{1}\\ M_{k}+1,if\;\left|\frac{d\alpha_{k}}{dt}\right|\leq \epsilon_{2} \end{array}\right.$$
where $\epsilon_{1}$ and $\epsilon_{2}$ are adaptation thresholds; $\left|\frac{d\alpha_{k}}{dt}\right|$ is the absolute value of the derivative of the signal; and $M_{k}$ is the current number of steps.

If none of the conditions (21) are met, the number of steps remains equal to $M_{k}$. The derivative $\frac{d\alpha_{k}}{dt}$ is approximated discreetly by(22)$$\frac{d\alpha_{k}}{dt}=\frac{\alpha_{k}-\alpha_{k-1}}{\Delta t}$$
where Δt is the time interval between steps.

This approach to determining *M* allows, on the one hand, to perform the adjustment according to the current state of the system, and on the other hand, balances the accuracy and speed of the system’s response and can be applied to a wide range of dynamical systems, including those with a physical pendulum. The threshold $\epsilon_{k}$ at step *k* is determined by smoothing the absolute value of the change in the signal $\left|\Delta \alpha_{k}\right|$, based on the Exponential Moving Average (EMA), i.e.,(23)$$\epsilon_{k}=\lambda \cdot \epsilon_{k-1}+\left(1-\lambda \right)\cdot \left|∆\alpha_{k}\right|$$
where $\epsilon_{k-1}$ is the threshold from the previous iteration; $\left|∆\alpha_{k}\right|=\left|\alpha_{k}-\alpha_{k-1}\right|$ is the absolute value of the change in the corresponding measurement signal in the current step; λ is a smoothing coefficient $\left(0<\lambda <1\right)$, which determines how quickly the threshold adapts to changes.

The thresholds $\epsilon_{1}$ and $\epsilon_{2}$ are set as relative multipliers of the current adaptive threshold:(24)$$\epsilon_{1}=\gamma \cdot \epsilon_{k};\;\epsilon_{2}=\delta \cdot \epsilon_{k}$$
where γ=1.5; δ=0.5.

The values of the coefficients *γ* and *δ* were selected based on simulation analysis and theoretical considerations. Simulations conducted on a system that replicated the dynamics of a physical pendulum showed that these values provide an optimal balance between sensitivity to rapid changes and stability in stationary states. Specifically, *γ* allows for rapid adaptation to abrupt changes, while *δ* ensures reliability in identifying stable states. The theoretical justification for the choice is based on the fact that the values set a symmetric interval around the current threshold $\epsilon_{k}$, which minimizes the risk of oversensitivity or underreaction.

The initial value of *M*, which is set in the algorithm, is M=10, and the initial threshold ϵ_k_ is defined as the average value of the first *M* changes.

This adaptive method offers a flexible mechanism for adjusting both the thresholds and the number of steps *M*, which ensures optimal performance of the measurement system under varying dynamic conditions.

### 3.4. Modeling and Tuning the Matrix Defining the Dispersions of the Model Errors Q

The model error covariance matrix **Q** in the Kalman filter characterizes imperfections in the theoretical model and the dynamic influences that cannot be predicted accurately. In the context of the current model, the errors described by the **Q** matrix can be divided into the following main categories: errors caused by approximations in the mathematical model; stochastic influences and dynamic conditions; errors related to the accuracy of mechanical parameters; and temperature and atmospheric influences.

Nonlinear components in the equations that describe the dynamics of a system can cause significant deviations, especially when the system experiences complex dynamic regimes that cannot be fully captured by a linearized model. At extreme values of the control impacts (such as large accelerations and angular velocities), deviations can occur because complex and dynamically changing conditions are not completely represented. These conditions include interacting inertial and centrifugal forces, vibrations, and external impacts that affect the system nonlinearly. All of these deviations, caused by the limitations of the approximations and stochastic influences, are reflected in the **Q** matrix, which allows for correction and adaptation of the model to changing conditions. Manufacturing deviations in mass and moments of inertia lead to differences between the values of these parameters set in the theoretical model and their actual values. These differences can lead to errors in calculating variables that describe the dynamic behavior of the pendulum in the model, such as its angular deviations and velocities, which are observed by the measurement system. Deviations in distances, such as the length of the pendulum or the distance $d_{z}$, can also cause inaccuracies in the predictions of the dynamic behavior of the pendulum.

For the accurate determination of the dispersions in the matrix **Q**, extended differential equations have been compiled that account for the complex dynamics of the system. These equations include nonlinear components that describe the dynamics of the system, the influence of inertial forces, and other significant factors that can cause deviations in the behavior of the system. Coriolis and centrifugal forces also have an impact as secondary effects that arise due to the movement of the object and cause additional forces and moments for the pendulum to deviate from the vertical.

Based on the above analysis of the dynamic influences and model limitations, the following differential equations have been derived:(25)$$\ddot{\alpha }+\frac{b_{1}}{I_{y}}\cdot \dot{\alpha }+\frac{m\cdot g\cdot l+m\cdot {\ddot{\zeta }}_{o}\cdot l+m\cdot l\cdot {\dot{\psi }}^{2}}{I_{y}}\cdot \alpha -\frac{m\cdot l\cdot {\dot{\alpha }}^{2}}{I_{y}}=-\frac{m\cdot l}{I_{y}}\cdot {\ddot{\eta }}_{o}-\frac{I_{y}+m\cdot l\cdot d_{z}}{I_{y}}\cdot \ddot{\theta }-\frac{2\cdot m\cdot l\cdot \dot{\alpha }\cdot \dot{\psi }}{I_{y}}$$(26)$$\ddot{\beta }+\frac{b_{2}}{I_{x}}\cdot \dot{\beta }+\frac{m\cdot g\cdot l+m\cdot {\ddot{\zeta }}_{o}\cdot l+m\cdot l\cdot {\dot{\theta }}^{2}}{I_{x}}\cdot \beta -\frac{m\cdot l\cdot {\dot{\beta }}^{2}}{I_{x}}=-\frac{m\cdot l}{I_{x}}\cdot {\ddot{\xi }}_{o}-\frac{I_{x}+m\cdot l\cdot d_{z}}{I_{x}}\cdot \ddot{\psi }-\frac{2\cdot m\cdot l\cdot \dot{\beta }\cdot \dot{\theta }}{I_{x}}$$
where *m*—mass of the pendulum; $I_{x}$, $I_{y}$—moments of inertia of the pendulum with respect to the axes $O_{1}y$ and $O_{1}x$, respectively; g—gravitational acceleration; *l*—length of the pendulum; ${\ddot{\eta }}_{o}$, ${\ddot{\xi }}_{o}$, ${\ddot{\zeta }}_{o}$—the longitudinal horizontal linear acceleration, transverse-horizontal linear acceleration, and vertical linear acceleration of the object, respectively; $2\cdot m\cdot l\cdot \dot{\alpha }\cdot \dot{\psi }$ and $2\cdot m\cdot l\cdot \dot{\beta }\cdot \dot{\theta }$ are the Coriolis forces; $m\cdot l\cdot {\dot{\psi }}^{2}\cdot \alpha$ и $m\cdot l\cdot {\dot{\theta }}^{2}\cdot \beta$ are the centrifugal forces.

The formulated differential equations described above serve as the basis for calculating the output data necessary to determine the dispersions in the matrix **Q**. These data are based on the measured values of the angular velocities and angular and linear accelerations in the corresponding iteration $\dot{\theta }\left(t\right);\;\ddot{\theta }\left(t\right);\;\dot{\psi }\left(t\right);\;\ddot{\psi }\left(t\right);\;{\ddot{\eta }}_{o}\left(t\right);\;{\ddot{\xi }}_{o}\left(t\right);\;{\ddot{\zeta }}_{o}\left(t\right)$, which are obtained from the AHRS mounted on the body of the mechanical module. Based on these measurements and the solution of Equations (25) and (26), the current values of $\alpha^{*}\left(t\right);\;{\dot{\alpha }}^{*}\left(t\right);\;\beta^{*}\left(t\right);\;{\dot{\beta }}^{*}\left(t\right)$ in step k−1 are calculated, denoted as $\alpha_{k-1}^{*};\;{\dot{\alpha }}_{k-1}^{*};\;\beta_{k-1}^{*};\;{\dot{\beta }}_{k-1}^{*}$. These values reflect the real system parameters calculated using the full nonlinear model.

When forming the matrix $Q_{k-1}$, it is necessary to take into account the differences between the values calculated with the extended model and the predicted values from the basic model of the Kalman filter. The index k−1 reflects the relationship of the matrix **Q** with the corresponding step in the iterative process in the filter, with the values being updated at each iteration according to the previous state.

The differences between the calculated and predicted values are used to assess the influence of the current dynamic conditions (such as vibrations, wind gusts, and other external impacts) on the system behavior.(27)$$∆\alpha_{k-1}=\alpha_{k-1}^{*}-\alpha_{k-1|k-2};\;∆{\dot{\alpha }}_{k-1}={\dot{\alpha }}_{k-1}^{*}-{\dot{\alpha }}_{k-1|k-2};\;∆\beta_{k-1}=\beta_{k-1}^{*}-\beta_{k-1|k-2};\;∆{\dot{\beta }}_{k-1}={\dot{\beta }}_{k-1}^{*}-{\dot{\beta }}_{k-1|k-2}$$
where $\alpha_{k-1}^{*},\;{\dot{\alpha }}_{k-1}^{*},\;\beta_{k-1}^{*}$, ${\dot{\beta }}_{k-1}^{*}$ are the values calculated in step *k* − 1 using the extended model, which takes into account additional nonlinear influences and instantaneous measurements; $\alpha_{k-1|k-2},\;{\dot{\alpha }}_{k-1|k-2},\;\beta_{k-1|k-2},\;{\dot{\beta }}_{k-1|k-2}$ are the predicted values from the basic model of the Kalman filter, using the information up to the previous step *k* − 2.

The differences from Formulas (27) determine the discrepancies between the predicted and measured values and have an important meaning for the accurate calculation of the dispersions that form the elements of the matrix $Q_{k-1}$. They allow the adaptation of the model to changing conditions and the accurate estimation of the errors in the system.

After calculating the deviations $∆\alpha_{k-1},\;∆{\dot{\alpha }}_{k-1},\;∆\beta_{k-1},\;∆{\dot{\beta }}_{k-1}$, their dispersions can be determined to form the matrix $Q_{k-1}$. The dispersions of these deviations are used to reflect the changing conditions and statistical characteristics of the errors in the model, and they play an important role in updating the Kalman filter. For this purpose, the successive values of the differences $∆\alpha_{k-1},\;∆{\dot{\alpha }}_{k-1},\;∆\beta_{k-1},\;∆{\dot{\beta }}_{k-1}$ are recorded for a certain interval of steps. This allows to collect sufficient information about the statistical dispersion of the deviations under different conditions. The average value of the deviations $∆{\bar{\alpha }}_{k-1},\;∆{\bar{\dot{\alpha }}}_{k-1},\;∆{\bar{\beta }}_{k-1},\;∆{\bar{\dot{\beta }}}_{k-1}$ is calculated using the following formulas:(28)$$∆{\bar{\alpha }}_{k-1}=\frac{1}{N}\cdot \sum_{i=1}^{N}∆\alpha_{k-1,i};\;∆{\bar{\dot{\alpha }}}_{k-1}=\frac{1}{N}\cdot \sum_{i=1}^{N}∆{\dot{\alpha }}_{k-1,i};\;∆{\bar{\beta }}_{k-1}=\frac{1}{N}\cdot \sum_{i=1}^{N}∆\beta_{k-1,i};\;∆{\bar{\dot{\beta }}}_{k-1}=\frac{1}{N}\cdot \sum_{i=1}^{N}∆{\dot{\beta }}_{k-1,i}$$

The dispersions of the deviations $\sigma_{∆\alpha_{k-1}}^{2},\;\sigma_{∆{\dot{\alpha }}_{k-1}}^{2},\;\sigma_{∆\beta_{k-1}}^{2},\;\sigma_{∆{\dot{\beta }}_{k-1}}^{2}$ are calculated to estimate the degree of dispersion around the average value. The formulas by which the dispersions are determined are(29)$$\sigma_{∆\alpha_{k-1}}^{2}=\frac{1}{N-1}\cdot \sum_{i=1}^{N}{\left(∆\alpha_{k-1,i}-∆{\bar{\alpha }}_{k-1}\right)}^{2};\;\sigma_{∆{\dot{\alpha }}_{k-1}}^{2}=\frac{1}{N-1}\cdot \sum_{i=1}^{N}{\left(∆{\dot{\alpha }}_{k-1,i}-∆{\bar{\dot{\alpha }}}_{k-1}\right)}^{2};\;\sigma_{∆\beta_{k-1}}^{2}=\frac{1}{N-1}\cdot \sum_{i=1}^{N}{\left(∆\beta_{k-1,i}-∆{\bar{\beta }}_{k-1}\right)}^{2};\;\sigma_{∆{\dot{\beta }}_{k-1}}^{2}=\frac{1}{N-1}\cdot \sum_{i=1}^{N}{\left(∆{\dot{\beta }}_{k-1,i}-∆{\bar{\dot{\beta }}}_{k-1}\right)}^{2}$$

The calculated dispersions allow updating the model to current conditions, representing the accuracy and statistical dispersion in the dynamic deviations. They are used to form the elements of the matrix $Q_{k-1}$, which has the following form:(30)$$Q_{k-1}=\left[\begin{array}{cccc} \sigma_{∆\alpha_{k-1}}^{2} & 0 & 0 & 0\\ 0 & \sigma_{∆{\dot{\alpha }}_{k-1}}^{2} & 0 & 0\\ 0 & 0 & \sigma_{∆\beta_{k-1}}^{2} & 0\\ 0 & 0 & 0 & \sigma_{∆{\dot{\beta }}_{k-1}}^{2} \end{array}\right]$$

This method of calculating dispersions provides flexibility and precision in updating the Kalman filter, ensuring that it can take into account dynamic changes and complex influences that cannot be fully predicted.

In addition, deviations in mechanical parameters such as mass, moments of inertia, etc., lead to systematic errors in the model, which must also be accounted for in the matrix $Q_{k-1}$. These deviations affect the formation of the final results of the model by creating systematic differences. For error estimation, associated with the accuracy of the mechanical parameters, the dispersion of the average value of the differences between the predicted values of the basic model of the Kalman filter (based on the information up to the previous step k−2) and the measured values from the AHRS mounted on the pendulum, $\alpha_{k-1}^{z_{1}},\;{\dot{\alpha }}_{k-1}^{z_{1}},\;\beta_{k-1}^{z_{1}},\;\beta_{k-1}^{z_{1}}$ in step k−1 was used.

The average values of the differences between the predicted and measured values at step k−1 characterize the systematic component of the error, which is a direct indicator of the impact of deviations in the mechanical parameters. The differences for the relevant parameters are defined as(31)$$∆\alpha_{k-1}^{mp}=\alpha_{k-1|k-2}-\alpha_{k-1}^{z_{1}};\;∆{\dot{\alpha }}_{k-1}^{mp}={\dot{\alpha }}_{k-1|k-2}-{\dot{\alpha }}_{k-1}^{z_{1}};\;∆\beta_{k-1}^{mp}=\beta_{k-1|k-2}-\beta_{k-1}^{z_{1}};\;∆{\dot{\beta }}_{k-1}^{mp}={\dot{\beta }}_{k-1|k-2}-{\dot{\beta }}_{k-1}^{z_{1}}$$
where $\alpha_{k-1}^{z_{1}},\;{\dot{\alpha }}_{k-1}^{z_{1}},\;\beta_{k-1}^{z_{1}},\;{\dot{\beta }}_{k-1}^{z_{1}}$ are the measured angles and angular velocities determining the pendulum deflection in the k−1st iteration.

The arithmetic average values of these differences for an interval of *N* steps are calculated by(32)$$∆{\bar{\alpha }}_{k-1}^{mp}=\frac{1}{N}\cdot \sum_{i=1}^{N}∆\alpha_{k-1,i}^{mp};\;∆{\bar{\dot{\alpha }}}_{k-1}^{mp}=\frac{1}{N}\cdot \sum_{i=1}^{N}∆{\dot{\alpha }}_{k-1,i}^{mp};\;∆{\bar{\beta }}_{k-1}^{mp}=\frac{1}{N}\cdot \sum_{i=1}^{N}∆\beta_{k-1,i}^{mp};\;∆{\bar{\dot{\beta }}}_{k-1}^{mp}=\frac{1}{N}\cdot \sum_{i=1}^{N}∆{\dot{\beta }}_{k-1,i}^{mp}$$

The dispersions of the differences from (31) will be equal to(33)$$\sigma_{∆\alpha_{k-1}^{mp}}^{2}=\frac{1}{N-1}\cdot \sum_{i=1}^{N}{\left(∆\alpha_{k-1,i}^{mp}-∆{\bar{\alpha }}_{k-1}^{mp}\right)}^{2};\;\sigma_{∆{\dot{\alpha }}_{k-1}^{mp}}^{2}=\frac{1}{N-1}\cdot \sum_{i=1}^{N}{\left(∆{\dot{\alpha }}_{k-1,i}^{mp}-∆{\bar{\dot{\alpha }}}_{k-1}^{mp}\right)}^{2};\;\sigma_{∆\beta_{k-1}^{mp}}^{2}=\frac{1}{N-1}\cdot \sum_{i=1}^{N}{\left(∆\beta_{k-1,i}^{mp}-∆{\bar{\beta }}_{k-1}^{mp}\right)}^{2};\;\sigma_{∆{\dot{\beta }}_{k-1}^{mp}}^{2}=\frac{1}{N-1}\cdot \sum_{i=1}^{N}{\left(∆{\dot{\beta }}_{k-1,i}^{mp}-∆{\bar{\dot{\beta }}}_{k-1}^{mp}\right)}^{2}$$

The dispersions of the arithmetic average values of the differences can be determined from the following expressions:(34)$$\sigma_{∆{\bar{\alpha }}_{k-1}^{mp}}^{2}=\frac{\sigma_{∆\alpha_{k-1}^{mp}}^{2}}{N},\sigma_{∆{\bar{\dot{\alpha }}}_{k-1}^{mp}}^{2}=\frac{\sigma_{∆{\dot{\alpha }}_{k-1}^{mp}}^{2}}{N},\sigma_{∆{\bar{\beta }}_{k-1}^{mp}}^{2}=\frac{\sigma_{∆\beta_{k-1}^{mp}}^{2}}{N},\sigma_{∆{\bar{\dot{\beta }}}_{k-1}^{mp}}^{2}=\frac{\sigma_{∆{\dot{\beta }}_{k-1}^{mp}}^{2}}{N}$$
where $\sigma_{∆{\bar{\alpha }}_{k-1}^{mp}}^{2},\sigma_{∆{\bar{\dot{\alpha }}}_{k-1}^{mp}}^{2},\sigma_{∆{\bar{\beta }}_{k-1}^{mp}}^{2},\sigma_{∆{\bar{\dot{\beta }}}_{k-1}^{mp}}^{2}$ are the dispersions of the arithmetic average values.

Then, the matrix $Q_{k-1}$, which takes into account the errors due to deviations in the mechanical parameters, will have the following form:(35)$$Q_{k-1}=\left[\begin{array}{cccc} \sigma_{∆\alpha_{k-1}}^{2}+\frac{\sigma_{∆\alpha_{k-1}^{mp}}^{2}}{N} & 0 & 0 & 0\\ 0 & \sigma_{∆{\dot{\alpha }}_{k-1}}^{2}+\frac{\sigma_{∆{\dot{\alpha }}_{k-1}^{mp}}^{2}}{N} & 0 & 0\\ 0 & 0 & \sigma_{∆\beta_{k-1}}^{2}+\frac{\sigma_{∆\beta_{k-1}^{mp}}^{2}}{N} & 0\\ 0 & 0 & 0 & \sigma_{∆{\dot{\beta }}_{k-1}}^{2}+\frac{\sigma_{∆{\dot{\beta }}_{k-1}^{mp}}^{2}}{N} \end{array}\right]$$

To ensure the accuracy and stability of the model, it is necessary to adapt the number of steps *N* used to calculate the dispersions of the deviations and angular velocities to the current dynamic conditions of the system. In this adaptive approach, two thresholds are used—a high threshold ($T_{hr}^{h}$) and a low threshold ($T_{hr}^{l}$)—which determine whether the number of steps should be increased, decreased, or maintained.

The thresholds are calculated based on the characteristic dispersions of the deviations under stable and unstable conditions. The system has been tested based on statistical processing of both real data and simulations, and it has been found that under stable conditions the dispersions of the deviations is around $\sigma_{base}^{2}$, which is why it is set to(36)$$T_{hr}^{h}=S_{1}\cdot \sigma_{base}^{2};$$(37)$$T_{hr}^{l}=S_{2}\cdot \sigma_{base}^{2}$$
where $S_{1}$ and $S_{2}$ are coefficients determined through simulations.

It was found that the optimal value is $S_{1}=2$, which indicates a doubling of the characteristic dispersion under unstable conditions, and $S_{2}=0.5$, which corresponds to half of the baseline dispersion under very stable conditions.

The base values of the dispersions for the angular deviations $\sigma_{∆\alpha_{k-1}}^{2}$, $\sigma_{∆\beta_{k-1}}^{2}$ and the angular velocities $\sigma_{∆{\dot{\alpha }}_{k-1}}^{2}$, $\sigma_{∆{\dot{\beta }}_{k-1}}^{2}$ are also determined based on statistical analysis of both real data and simulations and have the following values:$$\sigma_{base,angle}^{2}=1\times 10^{-4}\;{rad}^{2},$$$$\sigma_{base,angular\;velocity}^{2}=0.1225\;{\left(rad/s\right)}^{2}$$

The high threshold represents a limit beyond which the dispersion of deviations or angular velocities becomes significant, indicating unpredictable dynamic changes and fluctuations in the system caused by external factors or severe operating conditions. When the dispersion exceeds this threshold, the number of steps N is increased to provide greater averaging of the deviations, thereby enhancing the model’s stability. This approach helps mitigate the influence of random and short-lived anomalies.(38)$$\sigma_{∆\alpha_{k-1}}^{2},\sigma_{∆{\dot{\alpha }}_{k-1}}^{2},\sigma_{∆\beta_{k-1}}^{2},\sigma_{∆{\dot{\beta }}_{k-1}}^{2}>T_{hr}^{h}⟹N=N+∆N_{1}$$

Conversely, when the dispersion falls below the low threshold, it indicates that the system is operating under stable conditions. In such cases, the number of steps *N* is reduced, increasing the model’s sensitivity to future changes and enabling faster response to new deviations and dynamic effects.(39)$$\sigma_{∆\alpha_{k-1}}^{2},\sigma_{∆{\dot{\alpha }}_{k-1}}^{2},\sigma_{∆\beta_{k-1}}^{2},\sigma_{∆{\dot{\beta }}_{k-1}}^{2}>T_{hr}^{l}⟹N=N-∆N_{2}$$

If the dispersion is between the two thresholds, the number of steps is kept the same to maintain the current stability of the model, i.e.,(40)$$T_{hr}^{l}\leq \sigma_{∆\alpha_{k-1}}^{2},\sigma_{∆{\dot{\alpha }}_{k-1}}^{2},\sigma_{∆\beta_{k-1}}^{2},\sigma_{∆{\dot{\beta }}_{k-1}}^{2}\leq T_{hr}^{h}⟹N=N$$

This adaptive strategy enables the model to respond effectively to changing dynamic conditions while maintaining an optimal balance between stability and sensitivity. The algorithm begins with the initialization of the initial number of steps N=10. If the dispersions $\sigma_{∆\alpha_{k-1}}^{2},\;\sigma_{∆{\dot{\alpha }}_{k-1}}^{2},\;\sigma_{∆\beta_{k-1}}^{2},\;\sigma_{∆{\dot{\beta }}_{k-1}}^{2}$ exceed the threshold $T_{hr}^{h},$ the number of steps *N* is increased by $∆N_{1}=5$ to average out the fluctuations and achieve greater stability. This ensures a reduction in the impact of short-term anomalies. If the dispersion is below the low threshold ($T_{hr}^{l}$), which indicates that the system is operating under stable conditions, the number of steps *N* is reduced, for example, by $∆N_{2}=3$, to increase the model’s sensitivity to new changes. This adaptive strategy provides an optimal compromise between stability and model sensitivity to complex dynamic conditions.

## 4. Results

### Experimental Estimation of Dynamic Errors and Associated Uncertainty

The experimental studies were conducted to assess the limits and characteristics of the dynamic errors arising in the measurement process under various dynamic modes of object motion. Through the analysis of the associated uncertainty, the aim is to determine the reliability and accuracy of the developed prototype of the measurement system under conditions that closely resemble real ones.

To achieve this goal, experimental studies were conducted using a hexapod-type stand simulator, “Mistral” (Figure 6), which accurately reproduces complex angular and linear movements. The stand simulator provides high-precision simulation of movements in three linear coordinates (*x*, *y*, *z*) and three angular coordinates (*φ*, *θ*, *ψ*). The maximum linear movements that the hexapod can generate are ±250 mm, and the maximum angular oscillations reach ±30°. The repeatability of the hexapod plate’s movement has a deviation of less than 0.2 mm for linear coordinates and less than 0.04° for angular ones. The hexapod is pre-calibrated, with the expanded uncertainties of the reference reproduction of dynamic quantities being 0.5 mm for linear movements and 0.08° for angular movements. These metrological characteristics of the bench simulator ensure the achievement of the goals and objectives of the present study, which relate to the experimental assessment of the accuracy of the developed method and provide an opportunity for the validation of measurement systems in dynamic conditions. Additionally, the calibration of the hexapod ensures traceability of measurements to international standards and confirms the accuracy of reproducing the specified oscillations at different frequencies and amplitude modes.

The main quantity on which the dynamic accuracy analysis of the studied prototype was conducted is the dynamic error. This error is defined as the difference between the measured value $x_{meas}\left(t\right)$ obtained from the measurement system and the reference value $x_{ref}\left(t\right)$ from the reproduced movement data of the hexapod, i.e.,(41)$$e\left(t\right)=x_{meas}\left(t\right)-x_{ref}\left(t\right)$$

Dynamic error (41) includes both random and systematic deviations arising from noise and disturbances in the measurement system, dynamic influences such as vibrations and inertial forces, nonlinearities, and limitations in data processing algorithms.

Since the error from Equation (41) is a time-varying quantity, it must be evaluated using scalar metrological indicators, such as the mean value $\bar{e}$, which determines the systematic deviation, and the error dispersion $\sigma_{e}^{2}$, which characterizes the random variations; and the maximum error $max\left|e(t)\right|$.

The dynamic error is determined based on the measured and reference values at discrete moments in time. Therefore, Equation (41) is transformed into(42)$$e_{j}=x_{{meas}_{j}}-x_{{ref}_{j}},j=1,2,\ldots L$$
where $e_{j}$ is the error at time t=j; $x_{{meas}_{j}}$ is the measured value at time t=j; and $x_{{ref}_{j}}$ is the reference value at the same time.

The data discretization is performed with a step ∆t=10ms and is consistent with the update frequency of the data from the hexapod and the prototype measurement system. This sampling interval ensures proper synchronization of the recorded data and provides adequate time resolution for analyzing the dynamic accuracy of the measurement system.

In the present study, the accuracy assessment is carried out both through metrological indicators that characterize the error (mean value, variance, and maximum error) and through uncertainty theory, providing a complete metrological characterization of the accuracy of the measurement system. This combined approach allows, on the one hand, performing an analysis of deviations and, on the other hand, assessing the confidence in the results.

The expanded uncertainty U is defined by [45](43)$$U=k\cdot u_{c}$$
where *k* is the coverage factor, which for a 95% confidence interval is k=2, and $u_{c}$ is the combined uncertainty.

The combined uncertainty $u_{c}$ encompasses components that represent the main influencing factors on the calibration process and is determined by the following formula:(44)$$u_{c}=\sqrt{u_{meas}^{2}+u_{ae}+u_{rep}^{2}+u_{hex}^{2}+u_{axes}^{2}+u_{temp}^{2}+u_{nonlin}^{2}}$$
where $u_{meas}$ is the uncertainty associated with the difference between the measured and reference values; $u_{ae}$ is the uncertainty associated with the errors of the photoelectric absolute encoder; $u_{hex}$ is the uncertainty related to the accuracy of reproducing movements from the reference system; $u_{rep}$ is the uncertainty associated with the repeatability of the hexapod readings; $u_{axes}$ is the uncertainty due to the non-coincidence of the measurement axes; $u_{temp}$ is the uncertainty associated with temperature deviations; and $u_{nonlin}$ is the uncertainty arising from unforeseen complex conditions and nonlinear deviations.

The first component of the combined uncertainty $u_{meas}$ is determined by statistical analysis of the differences between the measured values $x_{{meas}_{j}}$ and the reference values $x_{{ref}_{j}}$. This uncertainty is classified as Type A, as it is derived from the processing of experimental data, i.e.,(45)$$u_{meas}=\sqrt{\frac{1}{L}\cdot \sum_{j=1}^{L}{\left(x_{{meas}_{j}}-x_{{ref}_{j}}\right)}^{2}}$$
where *L* is the number of measurements.

The $u_{meas}$ component includes most of the factors affecting the accuracy of the measurement system, including the influences of noise and other random disturbances.

The uncertainty associated with the errors of the photoelectric code converter, *u_ae_*, is determined based on the converter’s resolution, which is 0.082′. It is assumed that the converter’s errors lie within the bounds of two discrete values and are uniformly distributed. Therefore,(46)$$u_{ae}=b\cdot 0.082^{\prime }=0.098^{\prime }=0.0016{}^{\circ}$$
where *b*—coefficient that depends on the accepted type of distribution, and for a rectangular law b=0.6.

The determination of the component $u_{hex}$ related to the uncertainty of the reference system, is based on the previously stated expanded uncertainty at a coverage probability of 95%. Accordingly, $u_{hex}=0.25$ mm when reproducing linear movements and $u_{hex}=0.04{}^{\circ}$ when reproducing angular movements of the hexapod plate. The uncertainty component $u_{rep}$ associated with repeatability is determined in accordance with the declared Maximum Permissible Error (MPE) and the assumed rectangular distribution law, as expressed by the following equations:(47)$$u_{rep}=b\cdot {MPE}_{lin}=0.6\cdot 0.2=0.12\;mm-for\;linear\;coordinates$$(48)$$u_{rep}=b\cdot {MPE}_{ang}=0.6\cdot 0.04{}^{\circ}=0.024{}^{\circ}-for\;angular\;coordinates$$

The uncertainty component due to the misalignment of the measurement axes *u_axes_* is determined based on a simulation model, the 3D view of which is shown in Figure 6. The model includes the geometry of the hexapod and the mechanical module of the measurement system, along with all their main structural elements, allowing for accurate reproduction of geometric and mechanical interactions. The maximum angular deviation between the reference axes of the hexapod and the measurement system is estimated at 3°. Based on this limit value, various types of hexapod movements are simulated, and the MPE of the system readings is determined, the value of which is ${MPE}_{axes}=0.066{}^{\circ}$. A rectangular distribution law with a coefficient b=0.6 is again adopted. Therefore, the value of the uncertainty component due to misalignment of the measurement axes is $u_{axes}=b\cdot {MPE}_{axes}=0.6\cdot 0.066{}^{\circ}=0.04{}^{\circ}$.

The main influence on the uncertainty component associated with temperature deviations $u_{temp}$ is exerted by the sensors in the AHRS. Each of the sensors integrated into the AHRS—gyroscope, accelerometer, and magnetometer—is sensitive to temperature changes, leading to both systematic and random errors in their readings. Temperature variations can cause gyroscope drift, alter the output values of the accelerometer, and affect the magnetic field measured by the magnetometer. The component u_temp is determined by the declared *MPE*, which in this case is ${MPE}_{temp}=0.03{}^{\circ}$. A triangular distribution law is adopted, with the coefficient =0.42. Thus, the uncertainty component associated with temperature deviations will be equal to $u_{temp}=b\cdot {MPE}_{temp}=0.42\cdot 0.03{}^{\circ}=0.0126{}^{\circ}$.

The component of the combined uncertainty from unforeseen complex conditions and nonlinear deviations $u_{nonlin}$ is introduced due to the limited capabilities of experimental studies to cover all probable modes of motion of the moving object, especially those in extreme conditions. Nonlinear deviations arise from the inability to predict all dependencies between the motion parameters and the design characteristics of the measuring system, which cannot be accurately defined by linear models and are most evident in extreme conditions. In this work, the component $u_{nonlin}$ is determined through a combined analysis of simulation and experimental studies. The dynamic errors identified by the simulation $e_{sim}\left(t\right)$ and experimental $e_{exp}\left(t\right)=e\left(t\right)$ studies are expressed as time-dependent processes, and their differences $∆e\left(t\right)=e_{sim}\left(t\right)-e_{exp}\left(t\right)$ are analyzed for experimentally accessible modes. The values of $e_{exp}\left(t\right)$ are calculated based on Equations (41) and (42). The simulations are validated by comparison with experimental data in the same conditions and are then used to analyze dynamic deviations under conditions that cannot be investigated experimentally. The $u_{nonlin}$ component is determined based on the following formula:(49)$$u_{nonlin}=\sqrt{\sigma^{2}\left(∆e\right)+\sigma^{2}\left(e_{sim}^{ext}\right)}$$
where $\sigma \left(∆e\right)$ is derived from the differences between the simulation and experimental models, and $\sigma \left(e_{sim}^{ext}\right)$ is calculated from the dynamic deviations under extreme conditions.

In the present case, the component of the combined uncertainty from unforeseen complex conditions and nonlinear deviations is estimated at $u_{nonlin}=0.035{}^{\circ}$.

As mentioned above, the accuracy of the method proposed in this work is assessed through a comprehensive approach based on both error theory and uncertainty theory. To reduce the error from dephasing between the hexapod signals and those from the studied measurement system, necessary analyses were performed before conducting the experimental studies to achieve good synchronization of the processed signals. The hexapod control signals, which define the reference motion, and the data recorded by the measurement system were synchronized using a common time standard. For this purpose, an update frequency of 100 Hz was implemented, corresponding to the discretization *∆t* = 10 ms. Synchronization was achieved by using a control computer that sends commands to the hexapod in parallel while recording data from the measurement system. This method ensures that all data refer to the same moment in time, thus minimizing the dephasing between the measured and reference values. Additionally, tests were conducted to verify the time synchronization, examining the phase differences between the signals from the reference and measurement systems. The results show that the deviation between the signals does not exceed 1 ms, which is sufficient for an accurate study of the dynamic error.

To study the metrological indicators characterizing the error of the developed prototype of the measuring system based on this method, experimental studies were conducted in four operating modes. These modes encompass various real models, including characteristic and boundary conditions for the operation of the measuring system. Their detailed description is presented in Table 1. Photographic material from the studies is shown in Figure 7.

The results of the study of the measuring system presented in this work, under the purely harmonic motion of the hexapod plate with an amplitude of 10° and a frequency of 1 Hz, are given in Figure 8 in graphical form. The values of the metrological indicators that characterize the accuracy of the system under these conditions are as follows: maximum error $max\left|e_{ho}(t)\right|=3.9$′; average error ${\bar{e}}_{ho}=0.21$′; standard deviation $\sigma_{e_{ho}}=1.3$′. The measurement results with the proposed system and the reference signal describing the motion of the hexapod plate are shown in Figure 8a. The obtained results demonstrate a good correspondence between the measurements and the reference signal. The change in the dynamic error over time is presented in Figure 8b, and the histogram showing the distribution of the dynamic error values is depicted in Figure 8c. The distribution of the error values, according to the histogram in Figure 8c, is close to normal. This indicates that the influence of random factors on the errors is symmetrically distributed around the mean value, reflecting the stability of the system. The small standard deviation suggests that large deviations are rare and do not significantly impact the measurements. To conduct a comparative analysis of accuracy, studies were also carried out on a single AHRS under the same conditions, since as sensor is a key element in the system and provides a reference base for assessing the improvements achieved through the integration of a second AHRS and the application of a correction algorithm for dynamic errors. Graphs illustrating the results of the study of a single AHRS under the same conditions are shown in Figure 9. Figure 9a shows the correspondence between the measured signal and the reference signal, Figure 9b shows the dynamic error over time, and Figure 9c shows the histogram of the error distribution. The quantitative values of the error characteristics in this case are as follows: maximum error $max\left|e_{Aho}(t)\right|=4.6$′; average error ${\bar{e}}_{Aho}=0.25$′; standard deviation $\sigma_{e_{Aho}}=1.56$′. The results indicate that at low frequencies of oscillation of the moving object, the accuracy of the measurement system and the single AHRS is similar. This similarity is due to the reduced influence of inertial forces and moments, which are characteristic of more intense dynamic modes.

Unlike the studies in the previous mode, where the accuracy of the AHRS measurements and the presented measurement system were relatively close, significant differences in accuracy are observed in the study with a higher frequency, leading to almost twice the maximum error and an increased standard deviation in the AHRS. When studying the output signals from the AHRS in the sinusoidal motion mode of the hexapod with an amplitude of 5° and a frequency of 5 Hz, a tendency for signal “smoothing” is observed. The peaks and troughs of the output signal resemble a sinusoidal shape but are modulated at a lower frequency, with the signal’s amplitude varying and its phase shifting relative to the reference signal. This results from the limited frequency bandwidth of the AHRS and the influence of its built-in Kalman filter, which suppresses high-frequency components such as noise. These characteristics of the AHRS measurement signal are evident in Figure 10a, where the graphs of the measured signal and the reference signal from the moving plate of the hexapod are presented. The variation in the dynamic error over time is shown in Figure 10b. The maximum error in this case reaches a value of $max\left|e_{Ahf}(t)\right|=8.9$′, which emphasizes the limitations of the AHRS at higher frequencies. The histogram of the error distribution is presented in Figure 10c. The obtained value for the average error ${\bar{e}}_{Ahf}=0.29$′ indicates the presence of minimal systematic error, while the increased standard deviation $\sigma_{e_{Ahf}}=2.9$′ indicates a greater dispersion of the error values, associated with more intense dynamic conditions.

By the proposed method, which includes an advanced Kalman filter, the measurement accuracy in dynamic conditions significantly increases. The following values of the metrological characteristics were obtained in the conducted studies: maximum error $max\left|e_{hf}(t)\right|=4.4$′, average error ${\bar{e}}_{hf}=0.16$′ and standard deviation $\sigma_{e_{hf}}=1.46$′. These results are illustrated graphically in Figure 11a–c. The analysis shows that the system recovers both the amplitude and the phase of the signal, providing more accurate tracking of dynamic movements. The experiments conducted demonstrate that the developed system maintains an error that remains relatively constant regardless of the frequency of the reference signal. This emphasizes the robustness of the system at high frequencies, where more significant dynamic influences are observed.

Along with this, the system successfully suppresses unwanted modulated components that may occur at higher frequencies, providing stable and reliable measurements under high dynamic loads, which is evident when compared to the study of one AHRS. The advantage of the proposed methodology is particularly pronounced in comparison with standard approaches, where errors increase significantly with rising frequency. Thanks to the integration of the advanced Kalman filter, the system demonstrates high accuracy and adaptability, delivering precise results at both low and high frequencies of object movement.

The tendency to increase measurement accuracy with the system proposed in this work is also maintained when studying under conditions of added horizontal vibrations, the parameters of which are set in Table 1. The results show that while measuring under these conditions with AHRS, the maximum error is $max\left|e_{Ahv}(t)\right|=24.8^{\prime },$ the proposed system maintains an error of the order of $max\left|e_{hv}(t)\right|=6.9$′. This is also illustrated in Figure 12 and Figure 13, where the results of the study of the measuring system and one AHRS are presented in graphical form, respectively. In these conditions, vibrations are unwanted noise that can lead to deviations in measurements and reduce accuracy. The AHRS, using a built-in Kalman filter, suppresses high-frequency components but lacks a mechanism for distinguishing between local vibrations and real object movements. This results in additional errors when vibrations directly or indirectly affect the measured signals. Although the systematic error in this case does not reach large values, as illustrated by the average error ${\bar{e}}_{Ahv}=0.29$′, the dispersion of the error values is significant, since the mean squared value is $\sigma_{e_{Ahv}}=8.4$′.

The proposed system successfully compensates for these impacts thanks to its working principle, which includes the real-time correction of pendulum deviations. The mathematical model embedded in the Kalman filter accounts for the influence of linear and angular vibrations, integrating them into the system’s dynamic model. This approach allows the filter to calculate the effects of vibrations on the pendulum’s deflections and correct the measured signals in the measurement circuit. This correction eliminates errors caused by unwanted vibrations while preserving information about the actual movements of the object. All this is demonstrated by the graphs in the figures, which show the signals from the system and the reference signal, the change in dynamic error over time, and the distribution histogram, respectively. The histogram is a function of the average error, which in this case is ${\bar{e}}_{hv}=0.24$′ and the standard deviation $\sigma_{e_{hv}}=2.3$′.

The results of studies conducted under dynamic conditions simulating a random process also highlight the advantages of the measurement system over an AHRS in terms of measurement accuracy. A stationary random signal was used in the study because it is characterized by a mathematical expectation of zero and constant variance, making it suitable for simulating complex dynamic conditions across a wide range of frequencies. Due to their prevalence in technical applications under dynamic conditions, stationary random processes require specialized approaches for processing and analysis, especially in systems that utilize a built-in Kalman filter.

The reference signal defining the motion of the hexapod plate as a random process over time and the measured signal from an AHRS are presented in Figure 14a. The curves in the figure show that the AHRS can track the general shape of the reference motion but suppress the high-frequency components. This suppression is a consequence of the limited bandwidth and the built-in Kalman filter, which interprets these components as noise. The loss of high-frequency information results in significant deviations.

From the graph in Figure 14b of the change in time of the dynamic error, its behavior is clearly illustrated, with the maximum error reported being $max\left|e_{Ars}(t)\right|=28.8$′. The average error is ${\bar{e}}_{Ars}=0.15$′, and the standard deviation is $\sigma_{e_{Ars}}=9.9$′. The distribution of the dynamic error in the form of a histogram, which once again approximates a normal distribution, is presented in Figure 14c. Although in this distribution the errors are mainly clustered around the average error, the relatively large value of $\sigma_{e_{Ars}}$ indicates a greater variation in the errors, which leads to a higher probability of larger deviations from the reference signal.

For comparison with the previous analysis, the corresponding results of the experimental studies of the measurement system proposed in this work are presented in Figure 15a–c. The system shows significantly better metrological characteristics. The maximum error is limited to $max\left|e_{rs}(t)\right|=9.5$′ which is over three times lower than that of the AHRS. The average error is ${\bar{e}}_{rs}=0.68$′ and the standard deviation is $\sigma_{e_{rs}}=3.1$′. As shown in the results, the systematic component of the error in the proposed measurement system under random simulations is slightly larger than that of the AHRS. This increased value is due to the influence of the adaptive correction algorithm and the design features of the system. However, in the context of the total error values, this systematic component remains small and does not significantly affect the overall accuracy of the system. Moreover, the larger systematic component in the measurement system is an acceptable trade-off given the significant reduction in random errors and the minimization of the maximum error. Furthermore, as seen in Figure 15c, the errors are primarily clustered around the average value, while instances of maximum error are unlikely due to the low standard deviation. This indicates that the system is optimized to provide high accuracy under dynamic conditions while maintaining measurement reliability. These results show that the proposed system succeeds in preserving both high-frequency and low-frequency components of the signal while minimizing errors.

Based on the above study of the characteristics of accuracy in different dynamic operating modes, the component of the combined uncertainty $u_{meas}$, associated with the difference between the measured and reference values can be determined. The method of combining the standard deviation and the average error was used to determine $u_{meas}$, since it takes into account both random variations and systematic deviations. This approach provides a more balanced and objective assessment of uncertainty, especially in cases where the average dynamic error significantly influences the metrological accuracy of the system. The inclusion of these two quantities provides an adequate assessment that is better suited to the specifics of dynamic operating modes.

It is assumed that the errors are distributed according to the normal law, and the component $u_{meas}$ is calculated using the formula:(50)$$u_{meas}=\sqrt{{\left(b\cdot \sigma_{e_{rs}}\right)}^{2}+{\left({\bar{e}}_{rs}\right)}^{2}}$$

For a normal distribution with a 95% confidence interval =0.5, whence $u_{meas}=1.69$′=0.028°.

After substituting all the components in (44), the following value for the combined uncertainty is obtained: $u_{c}=0.065$°=3.9′. From this, it follows that the expanded uncertainty will be: $U=2\cdot u_{c}=0.13$°=7.8′.

Through the complex approach used in this work, which includes both analysis of error characteristics and detailed uncertainty assessment, a more complete metrological characteristic of the measurement system is provided. While error analysis offers information on the main metrological indicators, the maximum error, average error, and standard deviation uncertainty reflect the overall reliability and repeatability of the results. The combined consideration of both theories allows for a deeper analysis of the factors influencing the accuracy of the measurement while providing a means of quantifying the confidence in the results obtained. All this is of great importance in dynamic conditions, where many factors, primarily related to inertial impacts, can influence the measurement results.

## 5. Discussion

The proposed measurement system presents an alternative approach to determining the angular orientation of moving objects, based on the use of a physical pendulum as a reference element. Unlike conventional inertial systems based on stabilized sensors (e.g., high-precision gyroscopes), in this approach, the orientation in static conditions is determined based on the positional properties of the physical pendulum, which allows the formation of the reference vertical. This, together with the simplified design, results in low instrumental error, which is essential when initially setting up the site. In dynamic modes, the temporary deviations of the pendulum from the vertical, caused by inertial forces and moments, are compensated by two additional measuring circuits based on MEMS sensors, located both on the pendulum itself and on the system housing. The data from these two channels are processed by an algorithm of the Kalman filter, in the structure of which mechanisms for adaptive compensation of dynamic deviations and improved assessment of the real orientation of the object are implemented.

In order to verify the proposed measuring system, experimental studies have been carried out to assess its accuracy in dynamic mode. The results were compared with those of a standard inertial system of the AHRS type, used in the analysis due to its wide application as a standalone measuring tool for determining the angular orientation of moving objects. The error assessment of both systems was carried out by comparison with the reference movements reproduced by a hexapod reference system, which is calibrated in static and dynamic mode and provides traceability to international references.

The results obtained from the experimental studies are presented in a synthesized form in Table 2. It includes the maximum error, the average deviation value (as an indicator of systematic error), and the standard deviation, characterizing the accuracy in dynamic mode of both the proposed measurement system and the inertial system of the AHRS type. These values are calculated for four different operating modes, covering a wide range of real operating conditions.

In the first experimental mode, which is characterized by low-frequency harmonic motion, the measured angular orientation values from the proposed system and from the AHRS used show similar levels of accuracy. This result is expected because in this type of movement, the dynamic loads and inertial effects on the measuring system are weakly expressed.

Quantitative analysis of errors confirms this similarity. The maximum deviations are small, the mean error value of both systems is close to zero, and the values of the standard error are in a comparable range. The structure of errors over time shows relatively stable deviations without systematic drift, and the distributions of values are approximately symmetrical in nature around the center.

It is noteworthy that in both systems, larger deviations are observed near the extrema of harmonic motion—areas in which small phase shifts or amplitude inaccuracies can lead to distinct differences at low angular velocity. This feature is particularly important in precision applications where high precision is sought in all phases of movement.

The results of this mode show that both systems maintain good reproducibility and a low maximum error value with low dynamic impact. However, as seen in subsequent experimental modes, with increased dynamics and a more complex driving profile, the differences in error characteristics between systems become more pronounced.

In the second experimental mode, characterized by a higher frequency of movement, there is a distinct distinction in the behavior of the two measuring systems. While at lower frequencies the measurements from the AHRS showed acceptable comparability with the reference motion, here its accuracy deteriorates significantly. The measured signal demonstrates a smoothed shape, with impaired amplitude and phase response, which is the result of the limited bandwidth of the AHRS and the action of its built-in Kalman filter, configured with fixed covariance matrices. This type of setting, although stable under moderate dynamic changes, leads to a delayed response of the system at high frequencies and a sharp variability of input impacts.

It is in this mode that the efficiency of the proposed measurement system stands out, in which an adaptive structure of the Kalman filter is implemented. Unlike classical implementations with stationary covariance matrices, algorithms are used here to dynamically determine the covariances of both the model error and the measurement error. This allows the filter to adapt in real time to the current dynamic conditions—for example, in the event of sudden accelerations, changes in the driving profile, or fluctuations in the measured signals.

As a result of this adaptability, the measuring system maintains high accuracy even at higher frequencies. It is particularly important that the maximum error recorded in this mode remains commensurate with that recorded at a lower frequency, which clearly underlines the effectiveness of the dynamic adjustment of the covariance parameters. The quantitative characteristics confirm this advantage by significantly lower RMS error values compared to the AHRS values.

The results obtained demonstrate that the adaptive behavior of the filter is a key factor for the stability of the measuring system in conditions of increased dynamics, where traditional inertial systems with fixed parameters exhibit limitations.

In the third mode studied, horizontal vibrations were added to the main movement of the platform, which aims to simulate more real operating conditions. Under these conditions, sensor systems are subjected not only to a given movement but also to parasitic influences of a high-frequency nature, which can affect the accuracy of measurement.

The results of the analysis show a clear difference in the behavior of the two systems. The AHRS, due to its structure and built-in stationary Kalman filter, reacts to vibrations as noise, without the possibility of a clear distinction between real movements and local fluctuations. This leads to a significant scattering of the measured values and a violation of the form and phase of the measurement signal.

In contrast, the proposed measurement system demonstrates stable behavior, thanks to its design based on a physical model of the pendulum and the adaptive structure of the Kalman filter. The vibrations that act as disturbances on the mechanical system are detected through the additional sensors and integrated into the dynamic equation of the model. In this way, their impact on the results is neutralized already in the calculation process, without compromising the information about the real movements.

In the conditions of horizontal vibrations, the resistance of the developed system to parasitic influences is clearly visible. This is achieved thanks to the adaptive structure of the Kalman filter, which provides flexible tracking of changes in signal dynamics and effective compensation of both model errors and measurement noise.

When researched in the conditions of random influences, characterized by a wide frequency spectrum and lack of periodicity, the advantages of the proposed system stand out most clearly. To simulate this mode, a stationary random signal with a mathematical expectation equal to zero and a constant dispersion was used—a model that is widely used in the description of real technical impacts of a chaotic nature.

The comparative analysis performed shows that when using the AHRS, there is a tendency to suppress high-frequency components considered by the built-in Kalman noise filter. This leads to loss of information and significant deviations from real movement, especially in areas with sudden changes in movement. In this mode, the error has the largest amplitude and scattering compared to the other modes.

The proposed measurement system, on the other hand, demonstrates the stability of metrological characteristics thanks to the adaptive processing of input signals. The structure of the Kalman filter allows dynamic adaptation to changing signal statistics, successfully distinguishing actual movements from parasitic high-frequency components. In this way, sensitivity to real changes is preserved, without introducing additional distortions.

The results obtained confirm that even under the most unfavorable operating conditions, the system retains stability of behavior and the ability to reliably track the reference signal. Despite the higher error value compared to less intensive modes, there are no sharp deviations, loss of tracking, or deterioration of dynamic response, emphasizing the stability and adaptability of the measuring system in the presence of complex random impacts, typical of real practice.

The results of the experimental evaluation clearly demonstrate the advantages of the proposed measurement system over standard inertial solutions, especially in complex dynamic modes. The measuring system is characterized by high accuracy and stability in estimating angular orientation in a wide range of dynamic modes. This is achieved through an integrated correction circuit comprising two independent measuring channels, whose information enters the adaptive structure of the Kalman filter. Thanks to this architecture, effective compensation of dynamic errors is ensured by continuously adapting to changes in the model and measuring noise. The achieved results confirm the applicability of the developed system in technical areas requiring high reliability in real operation—including maritime navigation, autonomous mobile platforms, and industrial measurement solutions.

## 6. Conclusions

This work presents an innovative approach to measure the angular orientation of moving objects, based on real-time dynamic error correction rather than the stabilization of the inertial elements of the measuring instruments. Building on the principles of this method, an integrated measurement system has been developed that combines a simplified mechanical design with an advanced algorithmic platform for processing data from modern AHRS and MEMS sensors. The integration occurs at the system level as a functional union of a mechanical sensing element, sensor channels, and an adaptive algorithmic module. This architecture provides flexibility, adaptability, and traceability of measurements, allowing for modifications and metrological validation of individual components. These advantages are challenging to achieve with existing solutions based on complete hardware integration. By implementing a two-channel measurement model using two independent signals, high statistical reliability, and accuracy are attained in assessing the dynamic deviation of the pendulum from the vertical.

To increase the reliability and accuracy of the measurement system, especially for applications in vehicles with long-term routes, a module for correcting the systematic deviations of the sensor devices is incorporated into the system. This module is based on an additional algorithm that compensates for errors caused by long-term operation and variable driving conditions. An algorithm has been developed for adaptive estimation of the characteristics of the noises in the measurement channels and the errors allowed during measurement, and for adaptive estimation of the dispersions of the random quantities determining these deviations has been developed. In this way, optimal weighting of the measurement data and effective adaptation of the system to changing measurement conditions are ensured by the **R** matrix.

To adapt the theoretical model in the Kalman filter to the changing measurement conditions, an algorithm has been developed to determine the dispersions in the **Q** matrix, which corrects imperfections in the theoretical model related to the characteristics of the real measurement environment.

Based on experimental studies and a combined accuracy analysis rooted in both error theory and uncertainty theory, a metrological assessment of the system’s accuracy and reliability was conducted. The accuracy has been experimentally validated under both static and dynamic conditions, using a calibration reference system traceable to international standards, in accordance with ISO/IEC 17025 [46]. As a result, a metrologically sound methodology for dynamic calibration has been developed, providing a potential foundation for future practices and standardization in evaluating measurement systems for the angular orientation of moving objects under real dynamic loads.

## Figures and Tables

**Figure 1 sensors-25-04922-f001:**
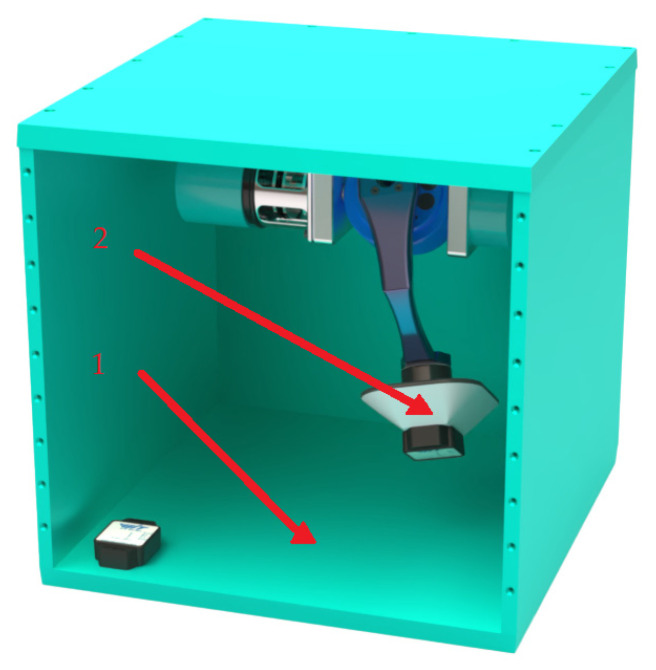
General view of the mechanical module of the measuring system. 1—Device body; 2—Physical pendulum.

**Figure 2 sensors-25-04922-f002:**
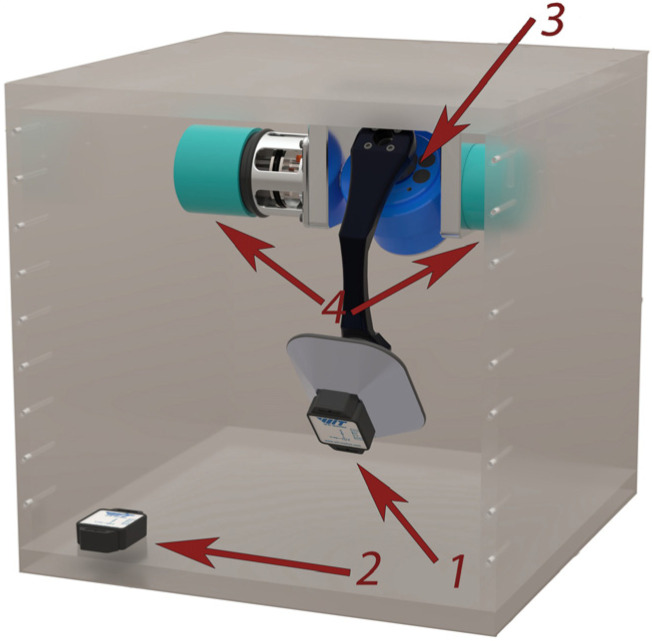
Detailed view of the cardan joint with mounted photoelectric absolute rotary encoders. 1—AHRS 1; 2—AHRS 2; 3—Cardan joint; 4—Photoelectric absolute rotary encoders.

**Figure 3 sensors-25-04922-f003:**
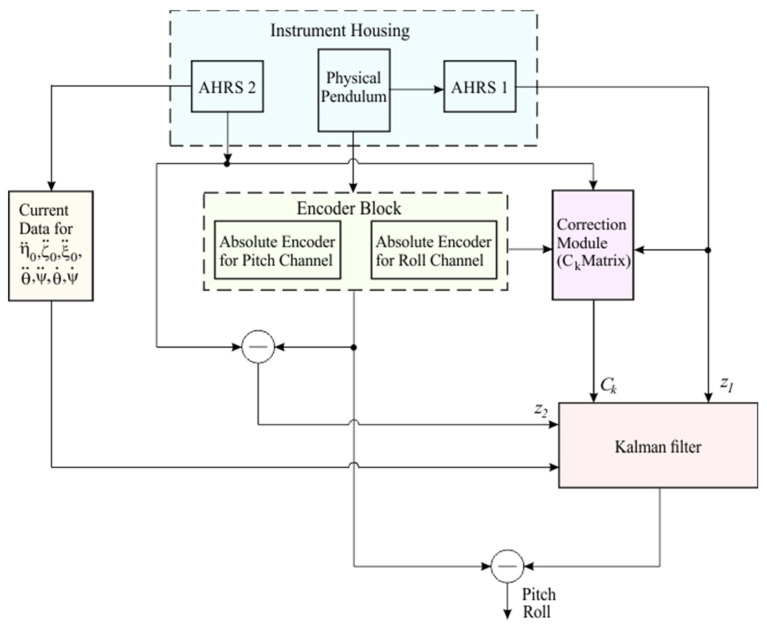
Structural diagram of the measurement system.

**Figure 4 sensors-25-04922-f004:**
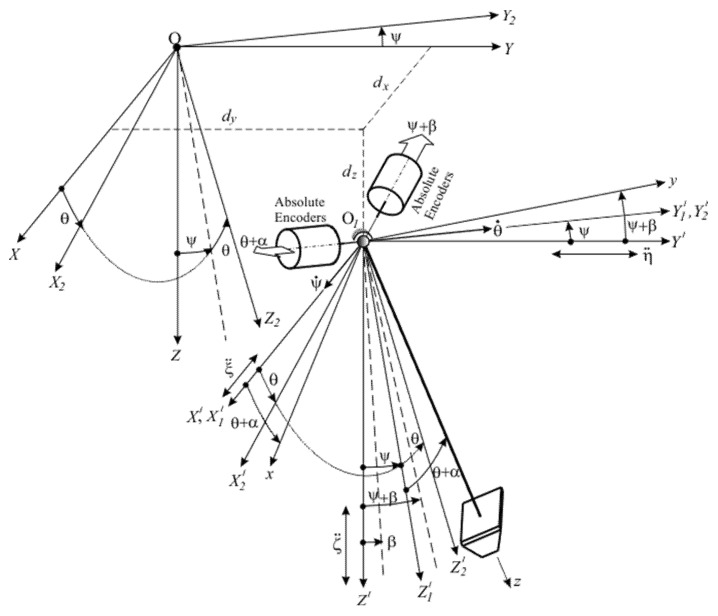
Coordinate systems defining the dynamic model of the measurement system.

**Figure 5 sensors-25-04922-f005:**
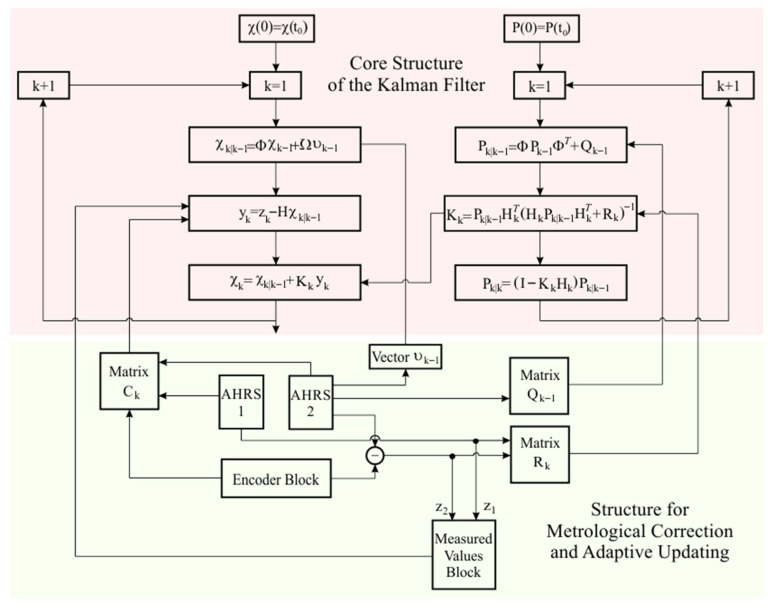
Structural diagram of the Kalman filter.

**Figure 6 sensors-25-04922-f006:**
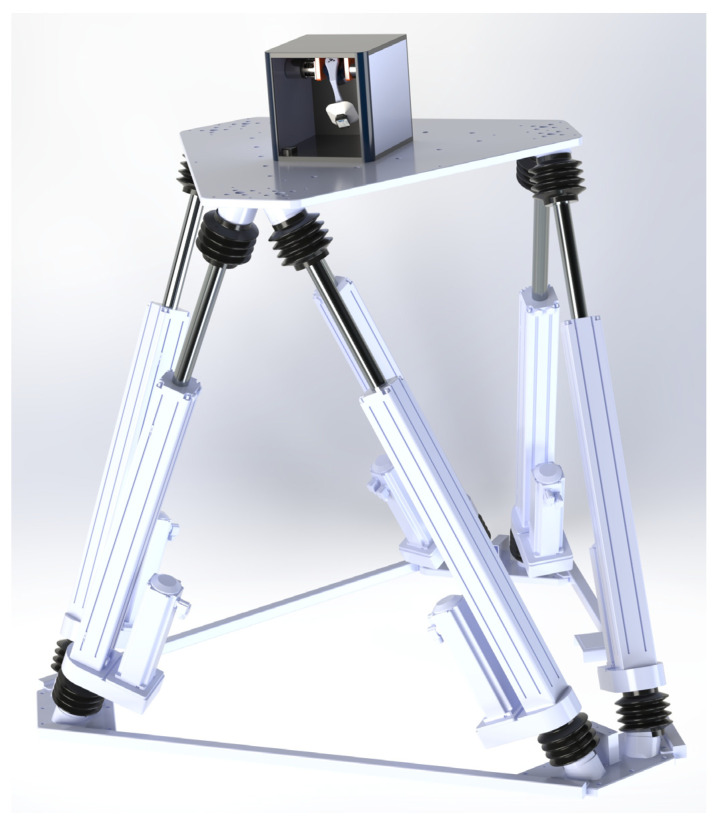
Three-dimensional model of the hexapod and the mechanical module of the measurement system.

**Figure 7 sensors-25-04922-f007:**
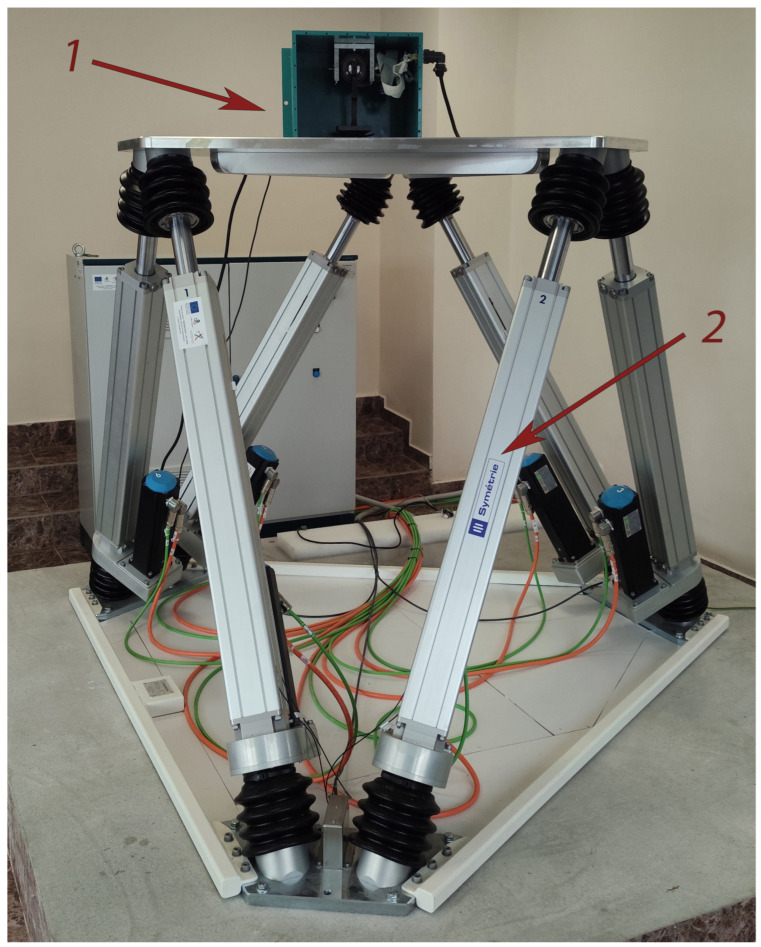
Photographic records from the experimental setup and procedures. 1—Hexapod; 2—Prototype of the measuring system.

**Figure 8 sensors-25-04922-f008:**
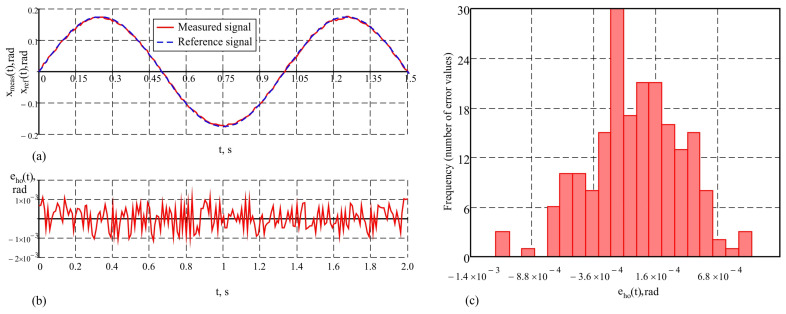
Graphical results from the study of the proposed measurement system under harmonic simulations: (**a**)—comparative analysis of the measured signal relative to the reference signal; (**b**)—variation in the dynamic error over time; (**c**)—histogram showing the distribution of dynamic error values.

**Figure 9 sensors-25-04922-f009:**
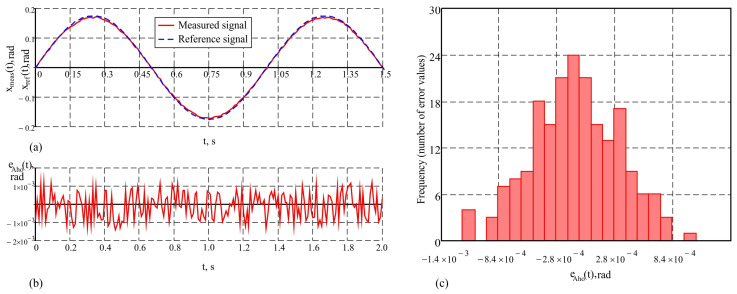
Graphical results from the study of the AHRS under harmonic simulations: (**a**)—comparative analysis of the measured signal relative to the reference signal; (**b**)—variation in the dynamic error over time; (**c**)—histogram showing the distribution of dynamic error values.

**Figure 10 sensors-25-04922-f010:**
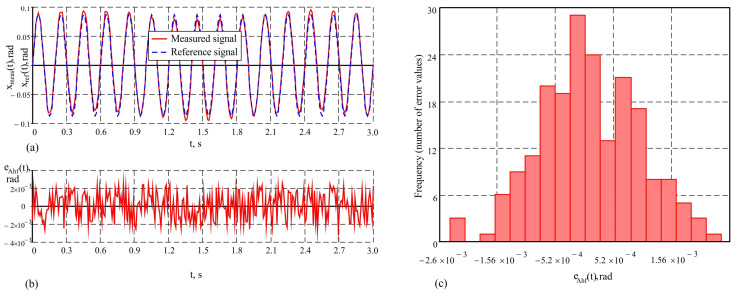
Graphical results from the AHRS study under sinusoidal simulation with an amplitude of 5° and a frequency of 5 Hz: (**a**)—comparative analysis of the measured signal relative to the reference signal; (**b**)—variation in the dynamic error over time; (**c**)—histogram showing the distribution of the dynamic error values.

**Figure 11 sensors-25-04922-f011:**
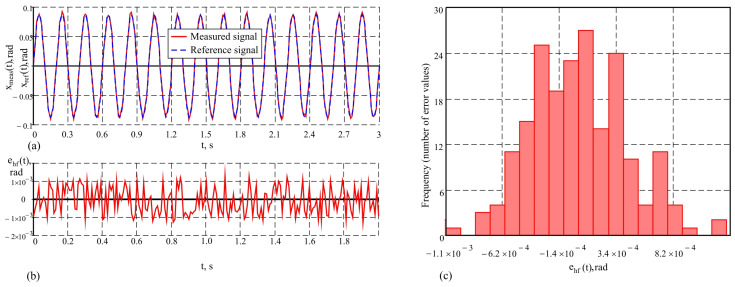
Graphical results from the study of the proposed measurement system under sinusoidal simulation with an amplitude of 5° and a frequency of 5 Hz: (**a**)—comparative analysis of the measured signal relative to the reference signal; (**b**)—variation in the dynamic error over time; (**c**)—histogram showing the distribution of the dynamic error values.

**Figure 12 sensors-25-04922-f012:**
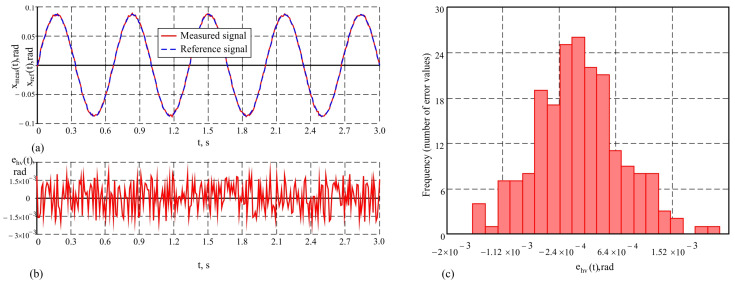
Graphical results from the study of the proposed measurement system under sinusoidal simulation with added horizontal vibrations: (**a**)—comparative analysis of the measured signal relative to the reference signal; (**b**)—variation in the dynamic error over time; (**c**)—histogram showing the distribution of dynamic error values.

**Figure 13 sensors-25-04922-f013:**
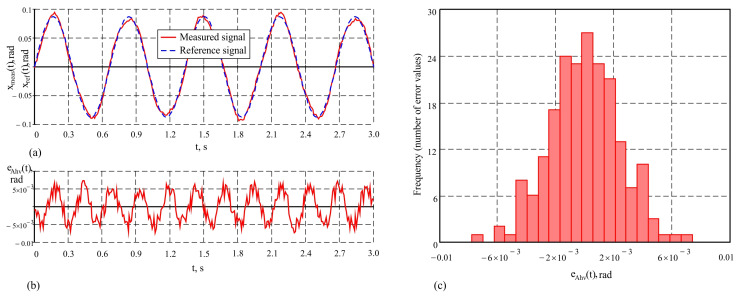
Graphical results from the AHRS study under sinusoidal simulations with added horizontal vibrations: (**a**)—comparative analysis of the measured signal relative to the reference signal; (**b**)—variation in the dynamic error over time; (**c**)—histogram showing the distribution of the dynamic error values.

**Figure 14 sensors-25-04922-f014:**
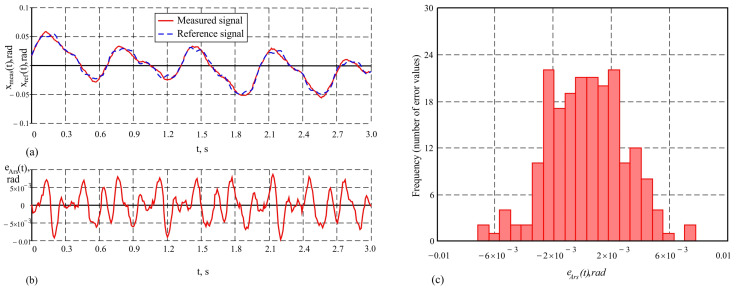
Graphical results of the AHRS study under the simulation of a stationary random process; (**a**)—analysis of the measured signal against the reference signal defining the motion of the hexapod plate; (**b**)—variation in the dynamic error over time; (**c**)—histogram showing the distribution of dynamic error values.

**Figure 15 sensors-25-04922-f015:**
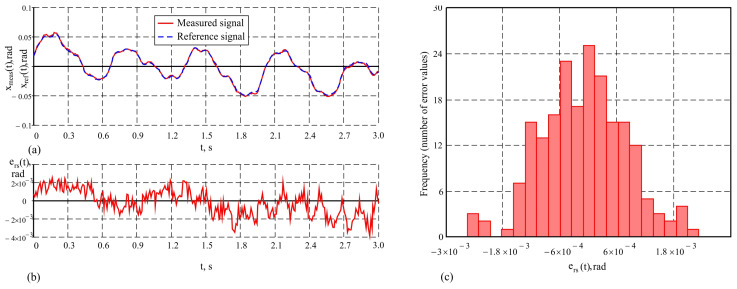
Graphical results of the measurement system study under the simulation of a stationary random process: (**a**)—analysis of the measured signal relative to the reference signal defining the motion of the hexapod plate; (**b**)—variation in the dynamic error over time; (**c**)—histogram showing the distribution of dynamic error values.

**Table 1 sensors-25-04922-t001:** Description of experimental modes for studying the metrological characteristics of the measuring system.

Operating Mode	Amplitude	Frequency	Description
Pure harmonic motion	10°	1 Hz	Sinusoidal motion, a basic test for system analysis under regular oscillations
Harmonic motion with high frequency	5°	5 Hz	This mode simulates angular oscillations of a moving object at a higher frequency. This item is to evaluate the system’s ability to track rapid changes in motion dynamics while maintaining measurement accuracy.
Harmonic motion with added horizontal vibrations	5° (main); 1 mm (horizontal vibrations)	1.5 Hz (main); 5 Hz (added vibrations)	Conditions are simulated where the system is subjected to external vibrations, as is often the case in real applications, such as the stabilizing of platforms onto vehicles, ships, or aircraft. High-frequency horizontal vibrations reflect the influence of mechanical disturbances that may impair measurement accuracy. This test allows the system’s resistance to such impacts to be assessed.
Movement in the form of a random signal	≤8°	≤5 Hz	Random movements are reproduced, typical of dynamic environments with abrupt disturbances, such as turbulences in aircraft, the effects of waves on sea vessels, or sudden changes in the trajectory of vehicles.

**Table 2 sensors-25-04922-t002:** Comparison of angular orientation estimation accuracy under dynamic conditions for the proposed measurement system and a conventional AHRS reference, across four motion scenarios.

Operating Mode	Maximum Error [Arcmin] Proposed/AHRS	Average Error Value [Arcmin] Proposed/AHRS	RMS Deviation [Arcmin] Proposed/AHRS
Pure Harmonic Motion	3.9/4.6	0.21/0.25	1.3/1.56
High-Frequency Harmonic Motion	4.4/8.9	0.16/0.29	1.46/2.9
Motion with Horizontal Vibrations	6.9/24.8	0.24/0.29	2.3/8.4
Random Signal Motion	9.5/28.8	0.68/0.15	3.1/9.9

## Data Availability

Data are available within the article.

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
