# Peer review of "A Method for Measuring Angular Orientation with Adaptive Compensation of Dynamic Errors"

_sensors, 2025, doi:10.3390/s25164922_

Round 1

Reviewer 1 Report (New Reviewer)

Comments and Suggestions for Authors
  1. In Page 4, the manuscript states that "the measuring system operates through an integrated method that combines a mechanical sensing element, two independent measuring channels with MEMS sensors, and an advanced Kalman filter." Could the authors clarify why a mechanical sensing element (physical pendulum) is necessary? Wouldn't a 6-axis MEMS sensor be a simpler and more compact solution?

  1. Figures 1 and 2 indicate that two AHRS units are employed in the measuring system. The author should provide the specific model numbers of these sensors and the reasons behind their selection?

  1. All figures in the manuscript appear to be of low resolution. The authors are advised to enhance the image quality for better clarity.

  1. The proposed measurement system demonstrates promising angular measurement accuracy. Could the authors provide a comparative analysis of its performance against other measurement systems, preferably presented in a tabular format?

  1. Regarding the adaptive characteristics of the proposed Kalman filter implementation, could the authors explicitly demonstrate where this adaptivity is manifested? A comparative analysis between conventional and adaptive Kalman filter results would be particularly insightful for readers.

Author Response

Dear reviewer,

We want to express our sincere gratitude for your valuable comments and remarks. Your observations have helped us significantly improve our work. Our responses to your comments are included below.

Reviewer comment:

In Page 4, the manuscript states that "the measuring system operates through an integrated method that combines a mechanical sensing element, two independent measuring channels with MEMS sensors, and an advanced Kalman filter." Could the authors clarify why a mechanical sensing element (physical pendulum) is necessary? Wouldn't a 6-axis MEMS sensor be a simpler and more compact solution?

Authors' response

We appreciate the reviewer’s insightful question.

The use of a mechanical sensing element – a physical pendulum – is not intended to replace modern MEMS sensors but is based on several essential arguments related to increasing the accuracy and metrological reliability of the system in real operating conditions.

Firstly, the physical pendulum allows the construction of a simplified mechanical structure with small instrumental error, which provides high accuracy in static and quasi-static operating modes. Due to its natural orientation along the gravitational vertical, the pendulum serves as a reference for the initial setting of the object, for example, when loading a vessel. In such situations, even small deviations in orientation – especially with a trim of several degrees – can significantly affect sailing performance and operational safety. In this context, it should be noted that in static mode, the measuring system achieves a maximum error of no more than one encoder discrete, which corresponds to 0.0014 degrees or 4.94 arcseconds. In this context, it should be noted that in static mode, the measuring system achieves a maximum error of no more than one encoder discrete, which corresponds to 0.0014 degrees or 4.94 arcseconds. This value substantiates the high level of accuracy essential for reliable angular orientation under initial and calibration conditions.

Second, the physical pendulum is used as an internal reference to identify systematic errors in MEMS sensors. In long-term operation (e.g. long sea voyages), drifts, errors due to temperature influences and instability of measurements may occur. Therefore, an adaptive mechanism for correcting systematic deviations has been implemented in the system, which uses the stable vertical position of the pendulum as a guide. (Section 3.2)

When the system finds, based on the data from the encoders, that the pendulum is in a stable vertical position, an assessment of the deviations between the current values supplied by the sensors and the zero values that define the reference vertical position of the pendulum shall be carried out. These differences are introduced into the Kalman filter by an adaptive correction matrix in the innovation signal equation. The procedure is implemented entirely software in the structure of the Kalman filter, without the need for additional hardware, thus enriching the functionality of the filter itself.

Third, the inclusion of the physical pendulum provides structural robustness and traceability in calibration and operational procedures, including dynamic modes, where stand-alone MEMS sensors, although combined in AHRS structures, often show sensitivity to internal drifts, temperature fluctuations and electromagnetic interference.

Unlike MEMS sensors, which do not have an independent physical reference, the physical pendulum provides a mechanical reference relative to the gravitational vertical that is not affected by electronic noise, temperature instability, or magnetic influences. It is this independence that underlies the mechanism for long-term self-assessment and correction of systematic errors.

Under the influence of inertial forces and deviation of the pendulum from the vertical, the system uses two independent correction chains, through which dynamic errors caused by the deviations of the pendulum from the vertical due to inertial forces and moments are compensated in real time.

Thus, the architecture of the system builds on standard AHRS solutions, adding the possibility of internal metrological verification and adaptive correction, impossible when using only inertial MEMS sensors.

Reviewer comment:

Figures 1 and 2 indicate that two AHRS units are employed in the measuring system. The author should provide the specific model numbers of these sensors and the reasons behind their selection?

Authors' response

We thank the reviewer for the remark made, which does point out our omission and the elimination of which will contribute to increasing the clarity and informative nature of the presentation. A text has been added to the article that indicates the model of the built-in AHRS devices. Reflected in L188 to L191

Reviewer comment:

All figures in the manuscript appear to be of low resolution. The authors are advised to enhance the image quality for better clarity.

Authors' response

All figures in the article are provided with improved resolution in the final version of the article to provide better visual clarity.

Reviewer comment:

The proposed measurement system demonstrates promising angular measurement accuracy. Could the authors provide a comparative analysis of its performance against other measurement systems, preferably presented in a tabular format?

Authors' response

We thank the reviewer for the relevant question and the recommendation made. In response to this, a new section entitled Discussion has been added to the article, which includes a comparative table with the results of an experimental study of the exact characteristics of the proposed system and of a standard inertial frame of type AHRS (Attitude and Heading Reference System). The data in the table are based on real measurements carried out under four different dynamic driving modes on the test platform (hexapod). Based on these results, a corresponding comparative analysis of the accuracy and stability of the two systems was carried out. (Table 2)

The study included a comparison with one of the widely used modern angular orientation measurement systems – AHRS, which combines MEMS gyroscopes, accelerometers, magnetometers and a built-in Kalman filter. This system is accepted as a reference due to its wide application as a stand-alone measuring instrument. It is important to emphasize that the AHRS used is not part of the system developed, but is an external tool used solely for the purpose of objective comparison.

The accuracy assessment was carried out in accordance with the requirements of the two modern metrological concepts – the theory of errors and the theory of uncertainty. The error theory analysis was carried out by comparison with the results of a reference measurement system – hexapod, which is calibrated in static and dynamic mode and provides traceability to international standards.

In addition, for an objective and reproducible assessment of accuracy, a specialized methodology for calibration in dynamic mode has been developed, through which the main components of the combined measurement uncertainty have been determined. This allows not only justified application of the system in real operating conditions but also provides end-users with a reliable tool for selection, implementation and certification of measuring systems of this type according to specific technical requirements.

Reviewer comment:

Regarding the adaptive characteristics of the proposed Kalman filter implementation, could the authors explicitly demonstrate where this adaptivity is manifested? A comparative analysis between conventional and adaptive Kalman filter results would be particularly insightful for readers.

Authors' response

We thank the reviewer for the extremely important question, which draws attention to one of the most important features of the proposed system – its adaptability. This question allows us to emphasize the essential role of adaptive mechanisms implemented in the structure of the Kalman filter in order to increase the accuracy and reliability of measurements in real dynamic conditions.

The purpose of adaptability in the proposed implementation of the Kalman filter is to provide sensitivity to variable operating modes, to take into account possible discrepancies between the mathematical model and the real system, as well as to improve the correspondence between the predicted and measured values in the presence of dynamic errors and noises.

Three adaptive algorithms have been implemented in the development, two of which are directly related to covariance matrices (the third for correction of systematic errors was discussed above):

  • Adaptive estimation of the covariance matrix of the model Q
    An algorithm for automatic evaluation of variances has been developed, which takes into account the imperfection in the mathematical description, as well as the influence of external disturbances. The method is based on observing the differences between predicted and real states, in which the filter adapts the degree of confidence to the model. This allows for a more accurate representation of dynamic and nonlinear processes that are not fully covered by the theoretical model.
  • Adaptive assessment of the covariance matrix of measurements R
    An adaptive approach for real-time estimation of measurement error variances is used, based on the analysis of the differences between the signals from the two independent measurement channels. Unlike classic fixed-value approaches, this method allows the filter to take into account the current characteristics of the noises and adapt its behavior to them.

In response to the reviewer's recommendation, a paragraph has been included in the newly created "Discussion" section that compares the efficiency of the adaptive structure of the Kallman filter with the classical implementation with stationary (fixed) matrices. The benchmarking is based on real-world experimental data and includes key metrics for accuracy in different dynamic modes. The results clearly show the advantage of adaptive structure, especially in conditions of vibration, acceleration and rapidly changing input influences. Reflected in L1114 to L1129

Reviewer 2 Report (New Reviewer)

Comments and Suggestions for Authors

This manuscript investigates the adaptive Kalman filter structure to achieve real-time correction and mitigate inertial element instability. Building upon this methodology, the study proposes an integrated approach for measuring the angular orientation of moving objects. Furthermore, it developed and implemented a MEMS sensor measurement system for determining roll and pitch angles, concurrently introducing a metrologically based quantitative evaluation method. This approach furnishes substantial insights pertinent to inertial measurement research. Nevertheless, numerous aspects remain incompletely understood or necessitate improvement. Please provide answers and corrections:

  1. The caption accompanying Figure 9 asserts that AHRS measurement accuracy is comparable to the proposed system. However, the figure demonstrates close alignment between the measurement signal and the reference signal, highlighting signal consistency that warrants explicit acknowledgment.
  2. Given that the reference signals in Figures 10 and 11 both possess an amplitude of 5° and a frequency of 5 Hz, the empirical basis for the conclusion (lines 932-935) stating "the developed system maintains an error that remains relatively constant regardless of the reference signal frequency" requiresclarification.
  3. The introductory section and theoretical analysis are excessivelyprotractedand should be concisely streamlined. Conversely, the experimental data presented in subsequent sections appears insufficiently robust and monotonous, necessitating a more balanced and comprehensive analysis.

4.The manuscript exhibits formatting irregularities requiring remediation: Figure serial numbers lack sufficient legibility and necessitate font enlargement and bolding. Suboptimal image placement results in excessive white space (e.g., Figures 1, 2, 6, 7). Redundant vertical margins are present in select figures (e.g., Figures 7, 12). Certain figures exceed appropriate dimensions and require proportional reduction (e.g., Figures 3, 4, 6). Additionally, comprehensive enhancement of image resolution is required. Table content must adopt lateral and vertical centering, as exemplified by Table 1.

Author Response

Dear reviewer,

We would like to thank you for your valuable comments. Our responses to these comments are included below.

Reviewer comment:

The caption accompanying Figure 9 asserts that AHRS measurement accuracy is comparable to the proposed system. However, the figure demonstrates close alignment between the measurement signal and the reference signal, highlighting signal consistency that warrants explicit acknowledgment.

Authors' response

We thank the reviewer for his comment. Indeed, a visual inspection of Figure 9a shows a significant overlap between the measured and the reference signal, which may give the impression of very high accuracy of the AHRS system.

We specify that Figure 9 presents the results of harmonic motion with a low frequency, in which the influence of inertial forces and moments on the measuring system is weak. This is the reason why similar accuracy is observed between the proposed system and the AHRS used, as noted in the text of the article.

In response to the remark, an additional section "Discussion" has been added to the article, which includes a comparative table with the quantitative values of the errors – both for the considered mode of harmonic motion and for the other experimentally studied modes. The table clearly illustrates the differences in maximum, mean and standard error between the proposed measurement system and the AHRS used, allowing for an objective assessment of the effectiveness of each (Table 2). In the same paragraph, an analysis of the observed values is made in order to present their interpretation from a metrological point of view.

In addition, to show the dynamic nature of the errors, a graph of error values over time is presented in Figure 9b, which gives a complete picture of the fluctuations and amplitude of errors at different time points.

Reviewer comment:

Given that the reference signals in Figures 10 and 11 both possess an amplitude of 5° and a frequency of 5 Hz, the empirical basis for the conclusion (lines 932-935) stating "the developed system maintains an error that remains relatively constant regardless of the reference signal frequency" requires clarification.

Authors' response

We thank the reviewer for his remark, which emphasizes the need for a more precise argumentation of the conclusion in “lines 932–935” (in the old version of the paper). This claim is based on a comparative analysis of the results obtained in two experimental modes with different input frequencies – 1 Hz and 5 Hz respectively. In these two modes, both the maximum and average error values of the proposed system remain in close values, indicating that there is no significant increase in frequency increase error.

We thank the reviewer for the constructive remark, which pointed us to the need for a clearer argumentation of the conclusion under consideration. As a result, an additional section "Discussion" was added to the article, in which a quantitative analysis of the errors observed in different dynamic modes was made. In this context, the role of adaptive covariance matrices for errors – both model and measurement – stands out, providing increased immunity and adaptability of the filter to changes in the frequency and nature of the input signal. Reflected in L1105 to L1129

It is important to emphasize that the selected frequencies are not arbitrary but are consistent with the characteristics of real objects where the system would be used – for example, marine vessels. At such objects, the frequency of roll and trim oscillations in real conditions usually does not exceed 5 Hz, which makes this frequency range completely adequate for the purposes of the study.

Reviewer comment:

The introductory section and theoretical analysis are excessivelyprotractedand should be concisely streamlined. Conversely, the experimental data presented in subsequent sections appears insufficiently robust and monotonous, necessitating a more balanced and comprehensive analysis.

Authors' response

We thank the reviewer for the valuable remark about the balance between the theoretical and experimental parts of the article. We agree that the presentation should be more dynamic and targeted. As a result of this recommendation, the introductory part was shortened in order to focus more clearly on the scientific novelty and contribution of the research, without unnecessarily lengthening the theoretical explanations.

Regarding the experimental part, it was expanded to include an additional section "Discussion", which presents comparative quantitative results under different dynamic modes, as well as an analysis of the effect of the adaptive algorithms used. In this way, a more balanced structure is provided between the theoretical and applied aspects of the development.

We hope that the improvements made will meet the expectations for clarity, conciseness and completeness of the exhibition.

Reviewer comment:

The manuscript exhibits formatting irregularities requiring remediation: Figure serial numbers lack sufficient legibility and necessitate font enlargement and bolding. Suboptimal image placement results in excessive white space (e.g., Figures 1, 2, 6, 7). Redundant vertical margins are present in select figures (e.g., Figures 7, 12). Certain figures exceed appropriate dimensions and require proportional reduction (e.g., Figures 3, 4, 6). Additionally, comprehensive enhancement of image resolution is required. Table content must adopt lateral and vertical centering, as exemplified by Table 1.

Authors' response

We thank the reviewer for the recommendations related to the layout of figures and tables. The following corrections have been made to the manuscript:

  • The numbering of the figures has been reworked with a larger and bolder font.
  • The layout of the shapes has been optimized to reduce the amount of empty space, including around Figures 1, 2, 6 and 7.
  • Redundant vertical margins in some figures (e.g. Fig. 7 and 12) have been removed.
  • The dimensions of the figures are uniform and, if necessary, reduced proportionally (Fig. 3, 4, 6).
  • The resolution of all images has been increased to ensure better visual perception.
  • The table contents, including Table 1, are centered both horizontally and vertically.

Round 2

Reviewer 1 Report (New Reviewer)

Comments and Suggestions for Authors

The author answered all my questions very well and made corresponding improvements to the manuscript. The manuscript can be accepted in present form.

This manuscript is a resubmission of an earlier submission. The following is a list of the peer review reports and author responses from that submission.

Round 1

Reviewer 1 Report

Comments and Suggestions for Authors

The authors proposed a method based on an advanced Kalman filter for angular orientation measurement of moving objects. The authors failed to show enough novelty and advantage of the proposed method compared to traditional techniques. And the organization and Engligh writing of the manuscript must be improved.

  1. There are too many mistakes in English writing. For example, ‘the world theory’in line 37, the non-English words in line 38-39, ‘[6] states’ in line 44, ‘.[8,9].’ in line 51, ‘The two photoelectric Absolute Encoders’ and so on.
  2. The authors claimed ‘an integrated method’in the title. However, it seems that the proposed design was quite different from the traditional on-chip-integrated sensors such as MEMS and MOEMS type. I don’t think it is a proper expression.
  3. They have given a too-long introduction in which many relative techniques have been mentioned. However, a distinct classification is still needed. And more references, especially review papers, are highly suggested to be added to give a better introduction. For example, as they mentioned MEMS-type sensors. There are much more other types than they have referenced.
  4. A proper conclusion of these traditional techniques with a quantitative comparison is suggested to be given when mentioned the limitation of traditional methods (in line 115-116).
  5. In line 152, the authors said ‘two photoelectric absolute encoders’are used. The Model and Company should be given in the manuscript if they are commercial sensors.
  6. The Kalman filters have already been used in relative sensors including displacement sensors, accelerometers and gyroscopes. The difference of the proposed model in this manuscript failed to show enough novelty and difference to other works, since the the authors mainly referenced Ref[34] to explain their model. More discussion is suggested to be added.

Reviewer 2 Report

Comments and Suggestions for Authors

The approach looks too complicated for implementation in real autonomous vehicles. The necessity of additional compensation measurements channels needs additional computational needs. Instead more researchers prefer to use additional correction via for example video observation and even GPS which may be accessible even in noisy and counteraction environment. It would be nice to pay attention to these possibilities. The Kalman filtering leads also to use the filtering with time dependent coefficients which must be taken from measurements with inevitable errors. This aspect is completely ignored in the article.